# Global gridded $NO_x$ emissions using TROPOMI observations

**Anthony Rey-Pommier[1,2,a], Alexandre Héraud[1], Frédéric Chevallier[1], Philippe Ciais[1,2], Theodoros Christoudias[2], Jonilda Kushta[2] and Jean Sciare[2].**

[1] Laboratoire des Sciences du Climat et de l'Environnement, LSCE/IPSL, CEA-CNRS-UVSQ, Université Paris-Saclay, 91190 Gif-sur-Yvette, France
[2] The Cyprus Institute, Climate and Atmosphere Research Center, 2121 Nicosia, Cyprus
[a] Now at: European Commission, Joint Research Centre, 21027 Ispra, Italy

**Correspondence:** Anthony Rey-Pommier (`anthony.rey-pommier@lsce.ipsl.fr`)

**Abstract.** We present top-down global gridded emissions of $NO_x$ for the year 2022. This dataset is constructed from retrievals of tropospheric vertical column densities of $NO_2$ by the TROPOMI spaceborne instrument associated with winds and atmospheric composition data from ECMWF reanalyses, using an improved version of a mass-balance atmospheric inversion. The dataset has a spatial resolution of $0.0625° \times 0.0625°$, and delivers a detailed overview of the distribution of emissions. It allows the identification of intense area sources, such as cities, and isolated emitters, such as power plants or cement kilns, but does not correctly represent biomass burning. At global level, the emissions obtained are consistent with the EDGARv6.1 bottom-up inventory, although there are differences at regional level, particularly in emerging countries and countries with low observation densities. The emissions of the three largest emitting countries, China, the United States and India, are 6, 14 and 4% lower than EDGAR estimates. Uncertainties remain high, and a quantitative analysis of emissions over several averaging periods indicates that averaging emissions uniformly across the year may be sufficient to obtain estimates consistent with annual averages, in regions of the world with high retrieval densities. This dataset is designed to be updated with a low latency to help policymakers monitor emissions and implement energy savings and clean air quality policies. The data can be accessed at `https://doi.org/10.5281/zenodo.13758447` as monthly files (Rey-Pommier et al., 2025).

## 1 Introduction

Air pollution is one of the leading causes of premature death in the world. Public health policies, implemented at the scale of countries, regions or cities, often aim to reduce the exposure to several pollutants, such as nitrogen oxides ($NO_x = NO + NO_2$). Such mitigation plans therefore require a precise knowledge of the emitters, as well as a monitoring of their emission levels over time. Data on $NO_x$ emissions is therefore fundamental for monitoring the implementation of air quality policies. Besides, because $NO_x$ is mainly produced during the combustion of carbon fuels at high temperatures, such data can also be, in conjunction with $NO_x/CO_2$ ratios derived at the scale of industrial sectors and countries, a tool to measure progress towards carbon neutrality. Gridded emissions with high spatial and temporal resolution are therefore of great scientific and political value. Many of such datasets are emission inventories, i.e. bottom-up models in which emissions are calculated on the basis of known sectoral activities and allocated in time and space, combined with specific emission factors by sector and, possibly, by country. These inventories provide valuable information on long-term trends and large-scale emission budgets, but they suffer from several weaknesses. They hardly represent daily or weekly variations, their activity data may be outdated, and some sources may be misallocated or unknown, which is common in many developing countries. Besides, uncertainties surrounding rapidly changing emissions factors and the generally low temporal resolution of activity data limits, in certain circumstances, the realism of such bottom-up inventories. Finally, they have a data lag of at least three years, which limits their potential as monitoring tools.

In this context, increasing efforts have been made to overcome the weaknesses of the inventories in order to obtain independent emission datasets that are homogeneous from one country to another. Such datasets are of the top-down type: they use direct observations of pollution and result from the inversion of an atmospheric chemistry-transport model (CTM) in which these atmospheric observations are assimilated. The observation data may be in-situ measurements or satellite retrievals.

In previous studies, we used a method for detecting and quantifying $NO_x$ emissions from daily observations of $NO_2$ columns by the TROPOMI instrument, onboard the Sentinel 5P satellite. This method, developed for the countries of the Eastern Mediterranean and Middle East region, is based on a two-dimensional simplification of atmospheric chemistry and transport, and does not require the direct use of a full 3D chemistry-transport model. Here, we extend the emissions domain to the whole world for the year 2022, and provide a dataset of averaged $NO_x$ emissions at a resolution of $0.0625° \times 0.0625°$. We analyse the results by pinpointing emitters and distinguishing between point sources, generally corresponding to isolated industrial facilities, and diffuse/area sources, generally corresponding to megacities. We also compare the results with the bottom-up inventory EDGARv6.1 and assess their reliability using different average horizons.

This article is structured as follows: Section 2 details the method used throughout this study, its improvements and simplifications since its previous uses, and the input data in its implementation. Section 3 presents the global $NO_x$ emissions dataset and analyses the different types of emitters. It also compares the results obtained with the EDGARv6.1 bottom-up inventory, and analyses different time horizons for averaging daily emissions in order to obtain representative results. Section 4 analyses the applicability limits of the method and highlights sources of uncertainty.

# 2 Methods

## 2.1 Input data

### 2.1.1 TROPOMI $NO_2$ column densities

$NO_2$ can be observed from space with satellite instruments based on its strong absorption features in the 400–465 nm wavelength region (Vandaele et al., 1998). By comparing observed spectra with a reference spectrum, the amount of $NO_2$ in a portion of the atmosphere between the instrument and the surface can be derived. The TROPOspheric Monitoring Instrument (TROPOMI), onboard the European Space Agency's (ESA) Sentinel-5 Precursor (S-5P) satellite, is one of those instruments (Veefkind et al., 2012). This instrument has a large swath width ($\sim$2600 km), combined with the 15-day orbit cycle of the satellite, leading to a revisit time of one day for every point of the Earth in absence of clouds. Moreover, these daily measurements are always collected during the middle of the day, the satellite crossing the sunlit equator at around 13:30 local time (LT). The high spatial resolution of the instrument (up to $3.5 \times 5.5$ km$^2$ since 6 August 2019) allows observing fine-scale structures of $NO_2$ pollution, such as hotspots within medium-size cities or plumes from power plants and industrial facilities. Tropospheric vertical column densities (VCDs, or simply "columns") are provided after retrieval of total slant column densities using the Differential Optical Absorption Spectroscopy method (Platt et al., 2008). VCDs represent the integrated number of $NO_2$ molecules per surface unit between the surface and the tropopause at the corresponding vertical. An algorithm also supplies an air mass factor, which is the ratio between slant and vertical column densities. This factor is derived from the knowledge of many physical quantities such as the vertical distribution of the absorber but also the viewing angle and the albedo of the observed surface. It comprises a significant part of the uncertainty in $NO_2$ measurements (Boersma et al., 2004; Lorente et al., 2019), which becomes non-negligible in a polluted atmospehre. Each TROPOMI retrieval is also associated with a quality assurance value $q_a$, which ranges from 0 (no data) to 1 (high-quality data). We selected $NO_2$ retrievals with $q_a$ values greater than $q_{a,\lim} = 0.75$, which correspond to clear-sky conditions (Eskes et al., 2022). Here, we use TROPOMI $NO_2$ retrievals in 2022 (OFFL product using processor version 2.5.0, product version 2.3.1 and 2.4.0 before and after November 2022 respectively). To limit effects due to product of processor version changes, other years are not studied.

### 2.1.2 Meteorological and air composition fields

Horizontal wind is taken from the ERA5 data archive, provided by the European Centre for Medium-Range Weather Forecasts (ECMWF). Both components have a horizontal resolution of $0.25° \times 0.25°$ gridded on 37 vertical pressure levels (Hersbach et al., 2020). We vertically average wind fields using the first two vertical levels, at 975 and 1000 hPa, except for representing ground winds, for which the last level at 1000 hPa is used. ECMWF also produces a reanalysis for air composition, under the Copernicus Atmospheric Monitoring Service (CAMS). It provides analyses and forecasts for reactive gases, greenhouse gases and aerosols. These parameters are gridded on 25 vertical pressure levels with a horizontal resolution of $0.4° \times 0.4°$ and a temporal resolution of 3 hours (Huijnen et al., 2016). Here, concentrations of $NO_2$, NO, OH, as well as temperature, are taken from CAMS to represent chemical processes in our model. Fields are vertically averaged using the first two vertical levels, at 950 and 1000 hPa.

 **2.1.3 Elevation data**

For computing altitude gradients, we use the Global Multi-resolution Terrain Elevation Data (GMTED2010, Danielson and Gesch (2011)) in its 0.0625°×0.0625° resolution version provided by the TEMIS data portal (`https://temis.nl/data/gmted2010/`). This version is derived from the original higher-resolution GMTED product (available at 30, 15, and 7.5 arc-seconds) to conveniently match coarser spatial scales. Elevation data is re-gridded on the TROPOMI grid, before calculation of the corresponding gradient to derive a corrective "topography-wind" value that is detailed in Section 2.2.2.

## 2.2 The mass-balance inversion

### 2.2.1 Main principle

The flux-divergence method is a mass-balance inversion model calculating the emissions of a given trace gas from observations of the corresponding vertical tropospheric columns, which is particularly well suited to data with high spatial resolution. In the case of $NO_2$, this approach was pioneered by Beirle et al. (2019). It has subsequently been implemented differently by other researchers, in different circumstances under simplified forms or, on the contrary, more complex ones (Lama et al., 2020; Rey-Pommier et al., 2022; de Foy and Schauer, 2022; Sun, 2022). The flux-divergence method is based on the conservation of mass principle, which makes it possible to calculate emission densities at the pixel scale as a function of a transport term and a sink term. By noting $C$ the local concentration of $NO_2$ and $\mathbf{w} = (u, v, w)$ the mean wind at the time of measurement, the corresponding emissions $E_C$ are expressed as:

$$E_C = \frac{\partial C}{\partial t} + \text{div}(C\mathbf{w}) + S_C \tag{1}$$

Here $S_C$ is the sink term expressing the loss of $NO_2$ due to chemical reactions. Assuming that the vertical variations in concentration are small compared with the horizontal variations, and considering that most $NO_2$ remains confined close to the ground, the previous equation can be rewritten in terms of tropospheric columns $\Omega$, which enables, in steady state, the computation of emissions per surface area $E$, as:

$$E = \frac{\partial(\Omega u)}{\partial x} + \frac{\partial(\Omega v)}{\partial y} + S_\Omega \tag{2}$$

$S_\Omega$ is the sink term expressed by surface unit. $D = \frac{\partial(\Omega u)}{\partial x} + \frac{\partial(\Omega v)}{\partial y}$ is the horizontal advection (transport) term. The assumption of a stationary state and a pollution close to the ground means that the temporal and vertical dimensions of the problem can be ignored, resulting in a purely 2D calculation of emissions. The corresponding reduction in complexity means that inversions can be performed very quickly compared with the conventional use of full-fledged 3D CTMs and without *a priori* knowledge on emissions. While useful, these simplifications come with inherent uncertainties, the main sources of which being on the input tropospheric columns, wind direction and atmospheric composition. It must also be noted that far from strong and localized sources, the underlying assumptions of stationarity and pollution containment are no longer valid.

Finally, the $NO_2$ production can be converted into $NO_x$ emissions. Performing this conversion is accounting for the portion of $NO_x$, mainly emitted as NO, which is not converted into $NO_2$ by reaction with ozone. The reformation of NO by the photolysis of $NO_2$ during the day leads to an equilibrium between the two compounds. The ratio $\mathcal{L} = [NO_x]/[NO_2]$ usually varies between 1.2 and 1.4, depending on local conditions. $NO_x$ emissions are therefore calculated as:

$$E_{NO_x} = \mathcal{L}E \tag{3}$$

In most urbanized areas, daytime NO concentrations frequently exceed 20 ppb. Under such conditions, this ratio is stabilized in a few minutes (Graedel et al., 1976; Seinfeld and Pandis, 2006). As this time is shorter than the inter-mesh transport timescale, the impact of stabilization time on the overall emission composition can be justifiably ignored. Here, it is near emission sources that the stationary hypothesis may not be applicable, in which case the value of $\mathcal{L}$ could be significantly higher than 1.4. The implications of this neglect are discussed in Section 4.1.

### 2.2.2 Refined version

In order to consider only anthropogenic pollution located close to the ground, it is necessary to remove any signal of natural emissions from the tropospheric columns provided by TROPOMI. In the absence of anthropogenic sources, the $NO_2$ columns that are observed constitute a tropospheric background $\Omega_b$. At the global scale, this background is

mostly due to soil emissions in the lower troposphere (Yienger and Levy, 1995; Hoelzemann et al., 2004). In the upper troposphere, $NO_2$ sources include lightning, convective injection and downwelling from the stratosphere (Ehhalt et al., 1992). We remove that background by calculating the 1st tercile in a 200 pixel × 430 pixel zone around each pixel (along × across track, i.e. approx. 700 km × 2360 km). We assume that this zone is sufficiently large whatever the considered pixel so that this tercile corresponds to the typical local value for this background. The large across-track distance is chosen for two reasons. Firstly, it limits the influence of low-quality observations and intermediate pollution levels on the background estimate, which can inflate the estimation when performed on a smaller domain and prevent the detection of smaller emitters. Secondly, it minimizes the mixing of pixels with varying $NO_2$ backgrounds due to distinct climatic and geographical conditions, such as between arid and temperate regions. Such variations are less pronounced across-track than along-track. Besides, the overpass of Sentinel-5P around 13:00 LT, when diurnal $NO_x$ variations are minimal, limits the mixing of pixels with backgrounds corresponding to different local times. After calculation of the background, it is subtracted from the vertical tropospheric column density, and the resulting lower tropospheric vertical density $\Omega' = \Omega - \Omega_b$ is used in the flux-divergence method. Pixels with columns lower than the calculated background have a corrected column reduced to $\Omega' = 0$. Such assumption can be challenged above macro-regions for which soil emissions and wildfires result in high $NO_2$ values observed by TROPOMI. High tropospheric backgrounds can also arise from localized paths of long-range transport of reactive nitrogen (Zhai et al., 2024) or around shipping lanes, where exhaust emissions directly increase $NO_2$ levels and may also enhance lightning activity that produces additional $NO_x$ (Thornton et al., 2017). The neglect of such effects is highlighted in Section 4.1.

We represent the sink term $S_\Omega$ by considering only the chemical loss of $NO_2$ due to its reaction with the hydroxyl radical (OH). This reaction follows a first-order kinetics, and the sink term can be expressed as $S_\Omega = k_{OH+NO_2}[OH]\Omega'$ with $k_{OH+NO_2}$ the reaction rate whose value is given by Burkholder et al. (2020). This is equivalent to compute a mixed lifetime $\tau = 1/(k_{OH+NO_2}[OH])$. This lifetime generally ranges between 1 and 14 h in mid-latitude regions and reaches higher values in polar and subtropical zones. Two global maps of lifetimes for winter and summer, computed with this method, are added in the Supplementary Materials. In many studies, this quantity is kept uniform and constant in the use of the flux-divergence method (Beirle et al., 2019; de Foy and Schauer, 2022), because justified by a relatively small domain of interest. Here, a singularity of our version of the flux-divergence method is to account for the temporal variability of OH, which is primarily driven by the amount of UV radiation from the stratosphere, but also for its spatial variability, since OH can also be influenced by $NO_x$ through a non-linear relationship (Valin et al., 2011). In this respect, our sink term is heavily reliant on the $NO_x$ sources accounted for in CAMS data. Neglecting a source, or mis-estimating the order of magnitude of its $NO_x$ emissions, therefore results in a wrong OH field whose bias depends on the amplitude of the neglect. Similarly, the coarse resolution in CAMS data (0.4°×0.4°) can fail to represent the particular conditions within or downwind power plant plumes, leading to a wrong estimation of the real OH budget. We regard these effects as less pronounced compared to those that would result in representing a constant lifetime for $NO_2$ which oversimplifies and misrepresents temporal and spatial dynamics by representing all situations the same way, whether they represent emitters or not.

Additionally, systematic artifacts concerning advection processes were reported over regions with complex topographies, particularly when high tropospheric vertical column densities are observed over mountainous regions. These high values can hinder the identification and quantification of point sources, possibly due to inaccurate mean wind fields over mountains. A study by Sun (2022) shows that these patterns can also be caused by 3D transport effects which have been ignored in the simplified 2D approach which has been described so far. A "topography-wind" $V$ term can be introduced in Equation 3 in order to correct for this effect using ground wind $\mathbf{w}_g$, the topography gradient $\nabla z_0$, and an inverse scale height $X_e$ as follows:

$$V = X_e \Omega' \mathbf{w}_g \cdot \nabla z_0 \tag{4}$$

Here, we choose a uniform and constant value of $X_e = 0.3$ km$^{-1}$. Although this value is about one order of magnitude lower than that used by Beirle et al. (2023), it corresponds to the approximate mean inverse scale height calculated by Sun (2022) in which a variability for $X_e$ was allowed by fitting its value through linear regressions on the basis of selected observations. While we acknowledge that choosing a single value for $X_e$ is a simplification, we note that performing the fit of its value would require an arbitrary selection of the cells used for that fit. We therefore compute the following equation to estimate $NO_x$ emissions:

$$E_{NO_x} = \mathcal{L}(\frac{\partial(\Omega' u)}{\partial x} + \frac{\partial(\Omega' v)}{\partial y} + k_{OH+NO_2}[OH]\Omega' + X_e \Omega' \mathbf{w}_g \cdot \nabla z_0) \tag{5}$$

Following de Foy and Schauer (2022), we perform the calculation of derivatives directly on the original TROPOMI grid (along-track and across-track) to better handle pixels with low-quality or no data, resulting in lower discontinuities in

the calculated transport term. To do so, we re-grid the wind field on the TROPOMI grid and linearly interpolate the estimates at the satellite timestamp. We do the same for all other parameters that are concerned for the calculation of the sink term (concentrations of OH, NO and NO$_2$, and temperature). Emissions are thus calculated on the TROPOMI grid and are then re-gridded on a regular north-south/east-west grid with a 0.0625°×0.0625° resolution.

Finally, the accuracy of TROPOMI retrievals can be compromised by challenges in estimating the air mass factor or local effects, particularly in specific vertical distribution scenarios (Griffin et al., 2019; Lorente et al., 2019; Judd et al., 2020). The latest versions of TROPOMI (v2.x) showed VCD values higher than those of earlier versions (v1.x), with biases up to 40%, depending on pollution levels and seasonal variations (Van Geffen et al., 2022). Additionally, the chemistry-transport model TM5, which is integrated into the operational TROPOMI product, tends to underestimate pollution near the ground, while overestimating NO$_2$ concentrations at higher altitudes over the sea (Latsch et al., 2023; Rieß et al., 2023). To compensate for such effects, studies like Goldberg et al. (2022) or Beirle et al. (2023) corrected the used VCDs by changing the corresponding vertical sensitivity over emitters. In this study, we do not perform such adjustment, while recognizing it could constitute a further step in the improvement of our dataset. On Figure 1, we sum up the functioning of our method.

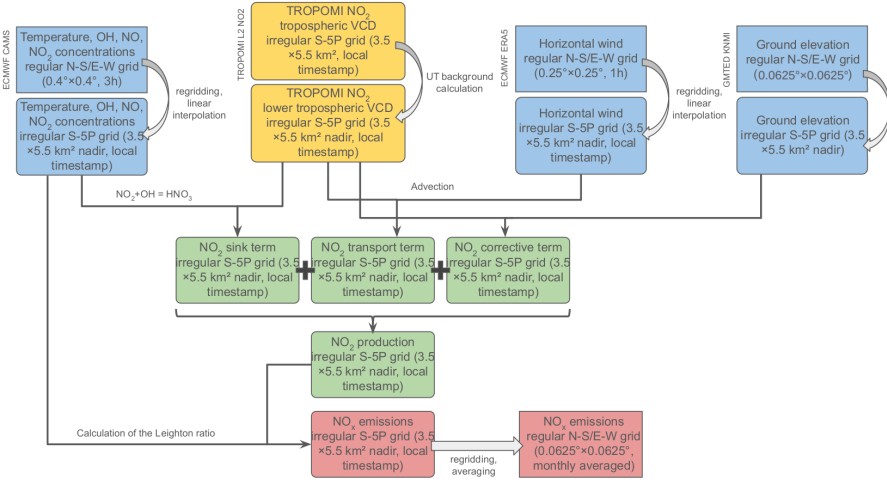

Figure 1: General overview of the mass-balance inversion.

## 2.3 EDGAR bottom-up inventory

Many high-resolution datasets for air quality exist at global (Benkovitz et al., 1996; Granier et al., 2019) or regional scale (Kuenen et al., 2022; He, 2012). Here we compare our averaged emissions for the year 2022 to NO$_x$ emissions provided by The Emissions Database for Global Atmospheric Research (EDGARv6.1) for 2018. It is a global inventory providing 0.1°×0.1° gridded emissions of greenhouse gases and air pollutants at the monthly scale, covering different sectors (Crippa et al., 2020). It is based on activity data of different nature (population, industrial processes, energy production, fossil fuel extraction, agricultural outputs, etc.) derived from the International Energy Agency (IEA) and the Food and Agriculture Organization (FAO), and the emission factors corresponding to each of the covered sectors. National and regional information on technology mix data provide a better characterization of these emission factors. End-of-pipe measurements are also used for correcting purposes. The version 6.1 of the inventory covers the years 1970-2018.

# 3 Technical validation

## 3.1 Spatial distribution of the global NO$_x$ emissions

The global map of the averaged NO$_x$ emissions for 2022 is shown on Figure 2, while Figure 3 zooms over seven macro-regions that cover most of the emitters over land and sea. Emissions are represented as density, i.e. by surface unit. All the analyses carried out in this study are based on the displayed domain, i.e. between latitudes 65°S and 65°N. This discards frequent outliers above these latitudes, resulting from monthly and annual estimates based on too few observations. Significant regional differences appear on these maps. The highest values are concentrated in developing areas such as eastern China, India and the Middle East. High values are also found in Europe, Russia and the United

States, where they correspond to megacities and industrial areas. Transport emissions can also be highlighted where they provide the highest share of emissions, i.e. on highways and shipping lanes which appear in various regions. South America, Oceania and sub-Saharan Africa display low or zero emissions except in a small number of cities and industrial sites. Wildfires, which are frequent in rainforests and savannas (Mebust and Cohen, 2013; Castellanos et al., 2014; Ossohou et al., 2019; Opacka et al., 2022), display quasi-zero emissions in Amazonia and low emissions in the Congo basin. Wildfire emissions might be under-estimated due to a wrong estimation of the lifetime, in particular in tropical regions where sinks other than the reaction with OH are important. Such other sinks are developed in Section 4.1. It must be noted that at a lower temporal scale, wildfire emissions display an annual variability. The example of the fires in the Congo basin is studied in the Supplementary Materials, with high emissions during summer (JJA). It is thus possible that a large number of smaller wildfires, occurring during other seasons, are too small to be correctly observed from space, as shown by other studies (Ramo et al., 2021; Khairoun et al., 2024).

Generally speaking, the maps highlights the industrialized areas, revealing the world's main megacities where several sources of emissions (traffic, power, residential) are mixed. Some industrial facilities and large power plants also appear. Emissions are correctly resolved in most regions of the world. The observed spread of emissions over two to three pixels (i.e. about 12 to 20 km) further away from the exact location of the corresponding emitters is due to the turbulent spread of emissions, which is not considered in our method. Finally, we note that emissions in mid- and high-latitude regions (beyond about 40° from the Equator) are noisy, due to an averaging over a smaller number of clear-sky days throughout the year. On average, countries such as Egypt, Niger and Saudi Arabia are observed about 70% of the time with a quality flag higher than $q_{a,\lim} = 0.75$, while Ireland, Canada and Finland are observed less than 20% of the time. This uneven sampling is also present in tropical regions where rainfall is frequent, as there is no measurement during cloudy scenes. Countries like Gabon, Indonesia or Peru are observed less than 30% of the year with quality flags higher than the threshold. In some cases, this low density of observations prevents emissions from intense sources from being quantified correctly at the monthly scale, as it is discussed in Section 3.4.

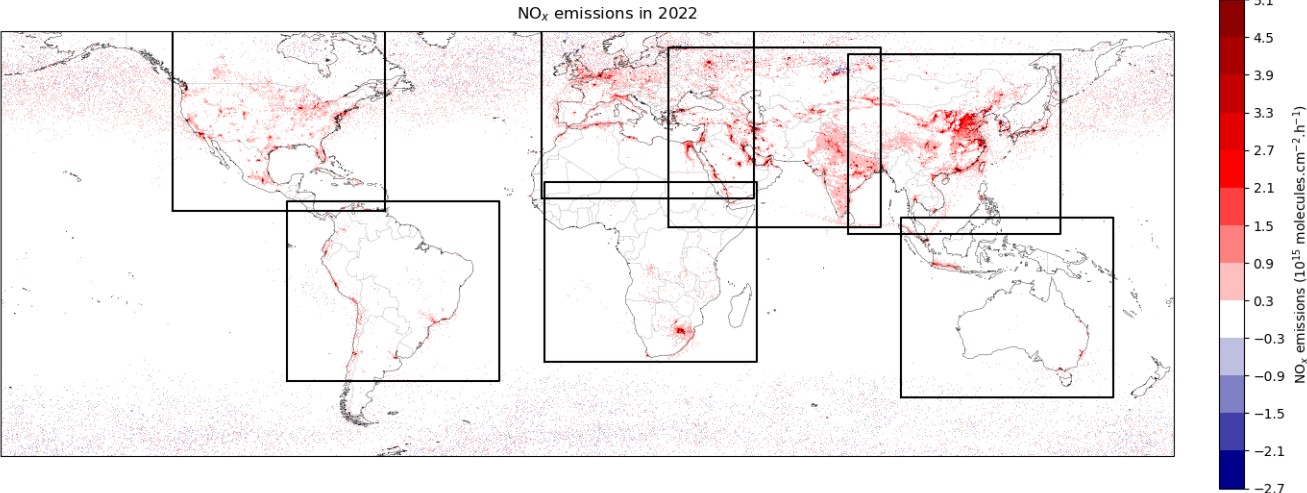

Figure 2: TROPOMI-derived mean daytime NO$_x$ emission rates in 2022 estimated with the flux-divergence method. The seven frames correspond to macro-regions whose emissions are specifically shown in Figure 3.

The statistical distribution of emissions is shown in Figure 4. Four different regimes of emissions can be distinguished in the red curve (note the log-log scale):

- Very low values of emission densities (less than ∼0.02 Pmolecules.cm$^{-2}$.h$^{-1}$), in practice at places where there are almost no emissions in reality. Note that, as the calculated fluxes represent averaged emissions, such pixels can also represent places where high emissions occurred, but only during a small portion of the year, as it is the case in regions where wildfires frequently happen.

- Residual emission densities (between ∼0.02 Pmolecules.cm$^{-2}$.h$^{-1}$ and ∼0.2 Pmolecules.cm$^{-2}$.h$^{-1}$), for which it is difficult to determine the corresponding source.

- Low emission densities (between ∼0.2 and ∼2 Pmolecules.cm$^{-2}$.h$^{-1}$), generally high enough to be associated with an emitter, but too low for a reliable quantification to be possible unless heavy averaging. The upper limit corresponds approximately to the emission densities observed on smaller power plants.

- High emission densities (higher than 2 Pmolecules.cm$^{-2}$.h$^{-1}$), where the signal-to-noise ratio is high enough to quantify emissions when enough observations are averaged.

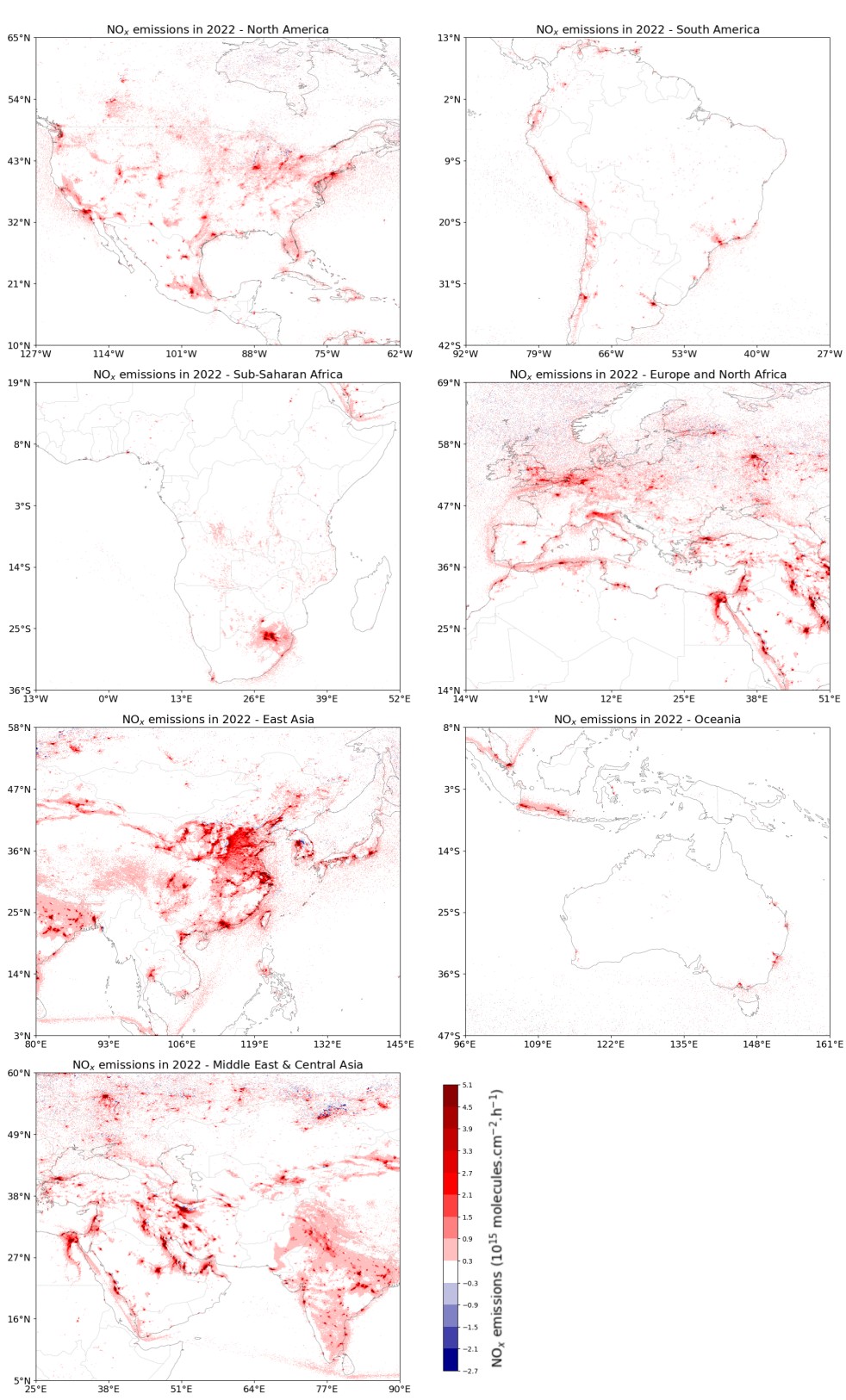

Figure 3: TROPOMI-derived mean daytime NO$_x$ emission rates in 2022 estimated with the flux-divergence method for North America, South America, sub-Saharan Africa, Europe and North Africa, East Asia, Oceania, Middle East and Central Asia.

Figure 4 also shows negative values (blue curve), even though negative emissions are physically impossible. They appear in practice because the transport term, which includes a derivative, can be negative. In calculated emission densities, negative pixels of low absolute value are as numerous as positive pixels of the same amplitude; they correspond to numerical noise and are found in pollution-free zones where the sink term is virtually zero. Higher values for negative pixels are less frequent: we count about 5 times less pixels with emission densities lower than -0.2 Pmolecules.cm$^{-2}$.h$^{-1}$ than pixels with emission densities higher than 0.2 Pmolecules.cm$^{-2}$.h$^{-1}$ (yellow and red parts of the graph in Figure 4). The locations where such high values are observed for negative pixels correspond to areas close to anthropogenic sources of NO$_x$, but in situations for which the absolute transport term has been overestimated or the sink term has been underestimated. Such negative emissions are limited to rare cases, such as Tehran, which is discussed in Section 4.2.

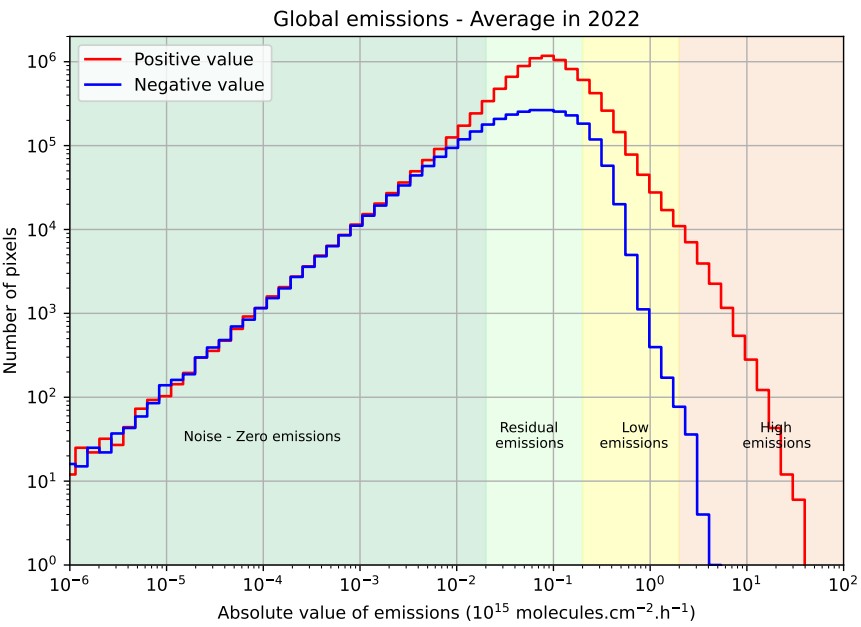

Figure 4: Distribution of positive and negative TROPOMI-inferred NO$_x$ emissions for year 2022. Four regimes can be distinguished (the values defining the thresholds between these regimes are given as order of magnitudes).

## 3.2 Diffuse sources and point sources

The assimilation of high-resolution observations with the flux-divergence method holds a significant potential for pinpointing emissions at small scale. As a consequence, it reveals the difference between sources that emit pollutants from a localized area, called point sources, from diffuse sources emitting pollutants over a wider area, such as sprawling urban regions like megacities. While the extent of the observed NO$_2$ pollution created by a point source is primarily determined by advection and turbulent mixing, the spread of the pollution for a diffuse source is above all determined by the spatial extent of the source itself. Point sources are therefore characterized by a dominance of the transport term, while diffuse sources (the term "area sources" is also used) exhibit a balance or dominance of the sink term (Beirle et al., 2019). Within the flux-divergence method, these two types of sources can be identified differently, since the main sources of uncertainty come from wind angle in the case of a point source while they come from the OH concentration explaining the sink term for a diffuse source. Because this distinction remains qualitative, to classify a detected source as one or the other type, arbitrary thresholds must be defined, concerning the number of pixels above a certain value of emissions, or the share of the transport term within the emissions in Equation 5. Here, we catalog all sources in the averaged emissions map for 2022. Firstly, we define a source as a cluster of at least 3 contiguous pixels above the value of 2 Pmolecules.cm$^{-2}$.h$^{-1}$. We then classify these sources as "point" or "diffuse" according to the number of pixels in the detected cluster, point sources being the clusters comprising 3 to 9 pixels, and diffuse sources those with more than 10 pixels. We detect 436 point sources and 323 diffuse sources, whose locations are displayed on Figure 5. The statistical distribution of the emitters, as well as their detailed location, are provided in the Supplementary Materials and in Rey-Pommier et al. (2025).

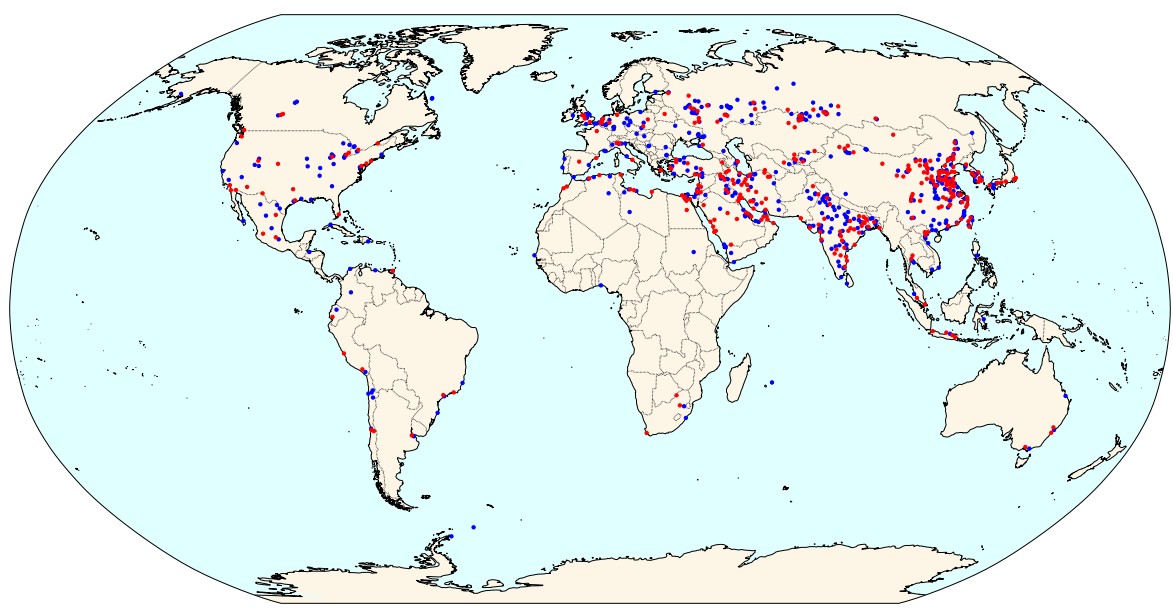

Figure 5: Location of different point sources in blue (between 3 and 9 contiguous pixels above 2 Pmolecules.cm$^{-2}$.h$^{-1}$) and diffuse sources in red (more than 10 contiguous pixels) for 2022.

### 3.2.1 Diffuse sources

Most point sources correspond to facilities such as power stations, cement kilns or mining sites. They can also correspond to concentrated urban areas. Conversely, most diffuse sources correspond to urban areas of megacities, whether they comprise industrial facilities within their extent or not. Exceptions concern mega-emitters like the Medupi and Matimba power plants in South Africa, mentioned in various articles (Reuter et al., 2019; Hakkarainen et al., 2021; Cusworth et al., 2023) or the Ain Sokhna industrial area in Egypt, already mentioned in Rey-Pommier et al. (2022). In both cases, such groups of industrial facilities exhibit particularly high emissions over more than 10 pixels and are detected as diffuse sources. Figure 6 displays the emissions of diffuse sources corresponding to megacities: Baghdad (32.3 t.h$^{-1}$, 198 pixels), Istanbul (15.4 t.h$^{-1}$, 127 pixels), Mexico City (17.6 t.h$^{-1}$, 114 pixels), Moscow (19.0 t.h$^{-1}$, 177 pixels), Riyadh (33.0 t.h$^{-1}$, 171 pixels) and Shanghai (100.2 t.h$^{-1}$, 836 pixels). City cores are denoted with dashed lines, and generally correspond to areas where emissions are largely above the cluster-detection threshold. Table 1 shows the 20 diffuse sources with the highest emissions.

| Number of pixels in cluster | Latitude (°N) | Longitude (°E) | Mean emission density (Pmolecules.cm$^{-2}$.h$^{-1}$) | Output (t.h$^{-1}$) | Emitter |
|---|---|---|---|---|---|
| 2623 | 37.617 | 116.030 | 2.818 | 217.89 | Beijing urban area, China |
| 836 | 31.283 | 120.354 | 3.771 | 100.21 | Shanghai urban area, China |
| 443 | 35.557 | 51.321 | 6.939 | 93.05 | Tehran urban area, Iran |
| 554 | -26.407 | 28.738 | 4.217 | 77.79 | Gauteng coal region, South Africa |
| 417 | 22.796 | 113.626 | 3.691 | 52.78 | Shenzhen & Hong-Kong urban area, China |
| 364 | 29.648 | 31.129 | 4.064 | 47.84 | Cairo & Beni Suef urban area, Egypt |
| 302 | 29.585 | 47.872 | 4.496 | 43.95 | Kuwait City urban area, Kuwait |
| 171 | 24.650 | 46.791 | 5.708 | 33.01 | Riyadh urban area, Saudi Arabia |
| 198 | 32.775 | 44.301 | 5.223 | 32.36 | Baghdad urban area, Iraq |
| 255 | 41.124 | 123.005 | 4.281 | 30.60 | Anshan urban area, China |
| 347 | 39.339 | 110.659 | 2.933 | 29.31 | Ordos mining region, China |
| 169 | 25.251 | 55.348 | 4.790 | 27.23 | Dubai urban area, United Arab Emirates |
| 193 | 37.162 | 126.822 | 4.425 | 25.30 | Seoul urban area, South Korea |
| 157 | 32.583 | 51.602 | 4.796 | 23.62 | Ispahan urban and industrial area, Iran |
| 124 | 21.112 | 39.313 | 4.886 | 21.03 | Djeddah urban area, Saudi Arabia |
| 220 | 37.320 | 112.088 | 3.131 | 20.39 | Shanxi urban area, China |
| 177 | 55.706 | 37.508 | 5.121 | 19.02 | Moscow urban area, Russia |
| 101 | 24.120 | 82.744 | 5.461 | 18.73 | Jogi Chaura industrial zone, India |
| 158 | 39.327 | 106.809 | 4.116 | 18.72 | Wuhai/Hainan industrial zone, China |
| 83 | -12.183 | -76.853 | 6.101 | 18.41 | Lima urban area & Pachamac mines, Peru |

Table 1: List and location of the 20 diffuse sources with highest TROPOMI-inferred NO$_x$ emissions (expressed as NO$_2$), and corresponding size of the cluster and main sector responsible for the emissions.

These six diffuse sources differ greatly from one another: Baghdad, Mexico City and Riyadh are very dense and isolated, allowing their emissions to stand out from the rest of the hotspots, while Moscow and Istanbul are less dense, resulting in lower emission densities. The Shanghai urban area has a large spatial extent, and the associated cluster extends over an area much wider than the city limits. Finally, it should be noted that Moscow and Shanghai experience many cloudy days, resulting in a fairly low level of averaging, leading to numerical noise that is visible on the maps. Many industrial facilities near city centers do not have high emissions, possibly due to an irregular production throughout the year, with high-activity periods covered by clouds.

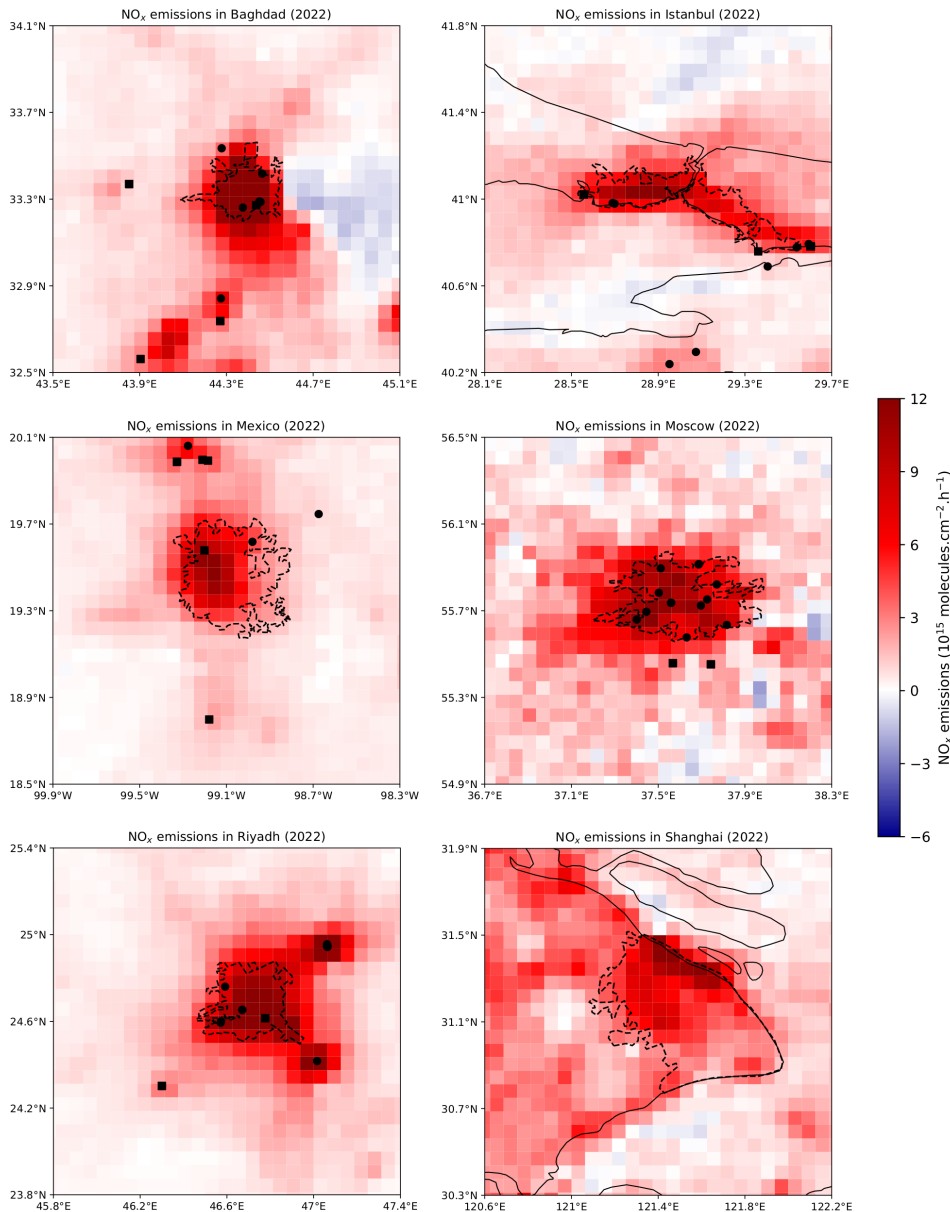

Figure 6: Map of mean daytime TROPOMI-inferred $NO_x$ emissions for 2022 (expressed as $NO_2$) for six megacities (diffuse sources), clockwise: Baghdad, Istanbul, Moscow, Shanghai, Riyadh, and Mexico City. The approximate boundaries of the city cores are denoted with dashed lines and the approximate location of power plants and other industrial facilities are denoted with circles and squares respectively, except for Shanghai (unavailable data).

### 3.2.2 Point sources

With a manual verification of the 436 detected point sources, we identify 48 outliers, 26 of which being points in places totally empty from any anthropogenic activity, and 22 points in areas with anthropogenic activity but without significant source (no facility of significant size). Most of these outliers are located in high-latitude regions, with 29 of them being located north to the 50°N parallel.

Because a threshold has been introduced to detect emitters, classified sources are isolated from each other. For many of them, emissions peak within the associated cluster. With a threshold set at 2 Pmolecules.cm$^{-2}$.h$^{-1}$, the corresponding signal-to-noise ratio is generally high enough to perform peak-fitting around the source, enabling accurate emission derivation. While this method works well for most point sources, it is not directly applicable to many diffuse sources. Since the observed spread of emissions around a source is caused by turbulent diffusion, a 2D-Gaussian function is applied to fit the detected sources within a $15 \times 15$ pixel zone around the maximum emission density within the cluster. Three examples are shown for the city of Medina, Saudi Arabia, the Sohar Industrial zone, Oman and the Western Mountain power plant, Libya on Figure 7. Note that these locations correspond to sources well-isolated from other industrial activities, in countries with frequent cloud-free conditions that allowed an averaging over high number of days in 2022.

We acknowledge the fact that the value of 2 Pmolecules.cm$^{-2}$.h$^{-1}$ (corresponding to 37 kg.h$^{-1}$ for a pixel at 60°N or 60°S, to 74 kg.h$^{-1}$ for a pixel at the equator) to mark the limit between high and low emissions is arbitrary, as other values for this threshold could be used. For instance, the Beijing cluster, identified on Table 1, with a size of 2623 pixels, is broken down into 35 smaller clusters (13 diffuse sources and 22 point sources) when changing the threshold from 2 Pmolecules.cm$^{-2}$.h$^{-1}$ to 3 Pmolecules.cm$^{-2}$.h$^{-1}$. These new clusters represent better urban sprawling around the various megacities and industrial facilities in Eastern China. However, in the same region, three point sources disappear when performing this threshold change. To determine the sensitivity of the point source and diffuse source detection and classification method, we carry out the detection by changing this threshold from 2 Pmolecules.cm$^{-2}$.h$^{-1}$ to 3 and 4 Pmolecules.cm$^{-2}$.h$^{-1}$. A comparative map is displayed in the Supplementary Materials. The point sources and diffuse sources are identified, and a fit with a 2D-Gaussian is carried out on point sources to estimate better emissions by accounting for the Gaussian nature of turbulent diffusion around the source. We then count the number of point sources with a fit of correct quality (with a correlation coefficient $R^2$ higher than 0.4). The results are shown in Table 2 for the different thresholds, and we compare the countries with the most point sources. Note that among the 48 outliers identified in the detected point sources with the threshold of 2 Pmolecules.cm$^{-2}$.h$^{-1}$, only 11 reached a value of $R^2$ higher than 0.4.

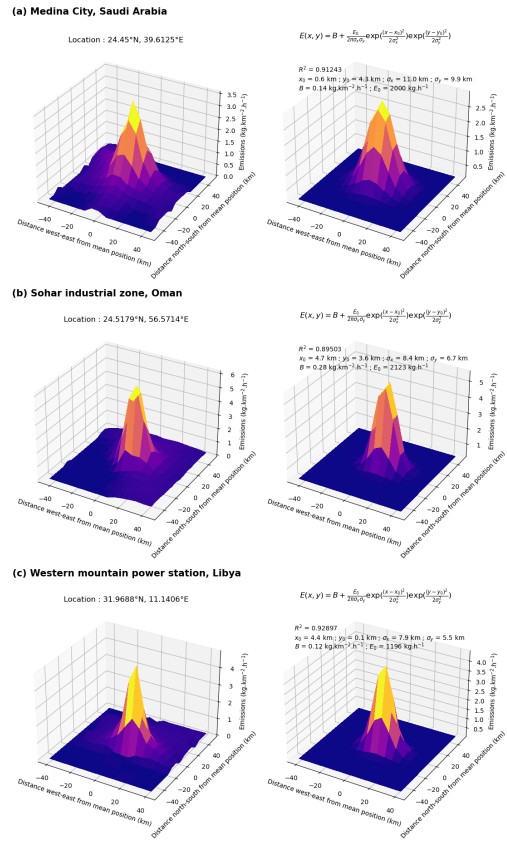

Figure 7: Calculated mean daytime NO$_x$ emissions in 2022 (expressed as NO$_2$) for different sources (left) and fitted emissions using a 2D-Gaussian function (right) for the city of Medina, Saudi Arabia (a), the Sohar Industrial Zone, Oman (b) and the Western Mountain power plant, Libya (c).

| Threshold value | 2 Pmolecules.cm$^{-2}$.h$^{-1}$ | 3 Pmolecules.cm$^{-2}$.h$^{-1}$ | 4 Pmolecules.cm$^{-2}$.h$^{-1}$ |
|---|---|---|---|
| **Number of point sources** | **436** | **287** | **163** |
| **Point sources with $R^2 > 0.4$** | **237** | **179** | **111** |
| China | 23 | 23 | 17 |
| India | 38 | 33 | 23 |
| Russia | 29 | 20 | 9 |
| United States | 17 | 4 | 3 |
| Türkiye | 8 | 5 | 2 |
| Iran | 7 | 8 | 9 |
| Saudi Arabia | 4 | 6 | 5 |
| Japan | 2 | 5 | 4 |
| Australia | 3 | 5 | 1 |
| Germany | 6 | 1 | 1 |
| Iraq | 6 | 4 | 2 |
| Mexico | 7 | 5 | 3 |
| Kazakhstan | 0 | 6 | 1 |

Table 2: Analysis of the number of point sources detected as a function of the threshold applied for cluster detection, and the number of point sources whose fit with a 2D-Gaussian was of acceptable quality ($R^2 > 0.4$). Countries with at least 5 point sources with one of the thresholds are displayed.

As seen with the example of Beijing, moving to a higher threshold can reduce the number of point sources by not including some emitters with lower emissions, but it can also increase the number of detected point sources by reducing the number of pixels corresponding to the cluster and moving certain emitters from the "diffuse source" category to the "point source" category. For example, with a limit of 2 Pmolecules.cm$^{-2}$.h$^{-1}$, the group of the Ras Laffan power stations in Qatar does not appear as a point source because its emissions are associated to a greater cluster corresponding to a diffuse source which includes the nearby Doha megacity. Conversely, with limits of 3 or 4 Pmolecules.cm$^{-2}$.h$^{-1}$, these power plants appear as a point source, and a good quality Gaussian fit provides their total emissions of 1.66 t.h$^{-1}$, close to the value of 1.86.h$^{-1}$ reported for the four-year average between 2019 and 2022 in Rey-Pommier et al. (2023). Finally, it should be noted that lowering the threshold to 1 Pmolecules.cm$^{-2}$.h$^{-1}$ also reduces the number of diffuse sources because several nearby urban areas become linked by residual emission zones into a single, larger, diffuse source. Conversely, lowering the threshold detects a very large number of point sources, but many of these additional points are outliers. In the rest of the study, we therefore choose to keep the lowest value of the threshold, i.e. 2 Pmolecules.cm$^{-2}$.h$^{-1}$, to optimise the number of correct emitters we work with. These emitters account for a total output of 2,303 t.h$^{-1}$ (352 t.h$^{-1}$ for point sources and 1951 t.h$^{-1}$ for diffuse sources). This represents about 17% of all emissions with densities higher than 0.2 Pmolecules.cm$^{-2}$.h$^{-1}$ (with a total output of 14,335 t.h$^{-1}$). As urban areas with more than 1 million inhabitants gather around 16% of the global population (Zimmer et al., 2023), this share of emissions from point and diffuse sources seems consistent with the detection limit of the flux-divergence method using TROPOMI retrievals, as urban areas lower than 1 million inhabitants are generally not detected as diffuse sources here.

The full list of the 436 point sources and 323 diffuse sources are given in Supplementary Materials. This list can be compared with the catalog provided by Beirle et al. (2023). Of the 237 point sources for which the Gaussian fit is of correct quality (with $R^2 > 0.4$), 137 also appear in their catalog. For these points, we generally obtain higher emissions (with a median of 441 t.h$^{-1}$ and an average of 487 t.h$^{-1}$ in our case, whereas they have a median of 303 t.h$^{-1}$ and an average of 353 t.h$^{-1}$). The two datasets have no particular reason to exhibit any clear correlation because they concern different years, and because their approach focused on monthly averages, while ours presents annual averages. For example, a site designated as a point source by Beirle et al. (2023) might not be detected if averaged over a whole year, especially if it stays inactive during certain periods. For instance, their catalog shows 187 occurrences where the signal of NO$_x$ emissions was significant for 6 months out of 12, and 348 occurrences for 5 months.

## 3.3 National and regional outputs and comparison with bottom-up emissions

We perform an analysis of emissions at the scale of countries by comparing them to the NO$_x$ emissions provided by EDGARv6.1 for 2018. For our TROPOMI-inferred emissions, we calculate the total mean NO$_x$ output, representing daytime emissions for 2022, for each country using country masks at the 0.0625°×0.0625° resolution. To avoid any over-estimation of the total output due to a very high number of pixels with very low emissions, we exclude from the calculation pixels with emission densities below 0.2 Pmolecules.cm$^{-2}$.h$^{-1}$. For emissions in EDGARv6.1, we sum the gridded emissions, representing monthly averages in 2018, for all sectors covered by the inventory and calculate the average flux for the year 2018. The output for each country is calculated using country masks at the 0.1°×0.1°

resolution. In both cases, we include pixels that directly touch coastlines because marine regions close to the shore receive the spread of anthropogenic emissions due to turbulent diffusion. This can result in over-estimating total emissions for smaller countries, especially those with low emission densities. In order not to account for such outliers, we exclude countries with a population lower than 300,000 inhabitants or with a size lower than 1,000 km$^2$ from our analysis. This concerns many insular countries in the Caribbean and the Pacific, as well as micro-states like Andorra or Singapore. Overseas territories are considered together with their mainland country. Figure 8 shows the country-wise comparison, covering 165 countries, and Table 3 provides a comparison at the scale of eight different macro-regions: Europe, North America & the Caribbean, South America, the Middle East & North Africa, former USSR countries, Oceania, sub-Saharan Africa and the rest of Asia. For each macro-region, differences are evaluated with the relative bias for the total region, and the mean absolute error (for which each country has the same weight). The use of these different metrics enables to assess the performance of the method on a large scale with respect to an inventory, while simultaneously evaluating its performance on a smaller scale to identify systematic effects that might offset each other at the larger scale.

TROPOMI-inferred emissions are generally close to EDGAR estimates for high-income countries, which generally have localized and powerful sources, or countries with a majority of sources located in areas with high observation densities. As a consequence, the macro-regions that perform best with both metrics are Europe, North America & the Caribbean, the Middle East & North Africa, and the rest of Asia. At the scale of countries, TROPOMI-inferred emissions are close to EDGAR estimates for the three largest emitting nations, i.e. China, the United States and India, with TROPOMI-inferred emissions 6, 14 and 4% lower than EDGAR estimates respectively. These three countries account for 45% of global estimated emissions. However, for the fourth highest emitting country, Russia, we estimate emissions 52% higher than in EDGAR. This difference can be due to the low density of observations for major emitters in Russia, leading to the estimation of monthly emissions on the basis of only a few estimates. To illustrate this, the monthly emissions of the two largest Russian cities, Moscow and Saint Petersburg, are studied in the Supplementary Materials. In extreme cases, such key emitters can have no estimates at all for months, making the calculation of the annual average representative of only a part of the year, even when its order of magnitude is correct. Generally speaking, large differences between our top-down estimates and EDGAR emissions are found for many countries that also have low observation densities for this reason. Without prior knowledge of the annual emission profiles in these countries, these biases cannot be corrected, leading to a systematic mis-estimation of total emissions.

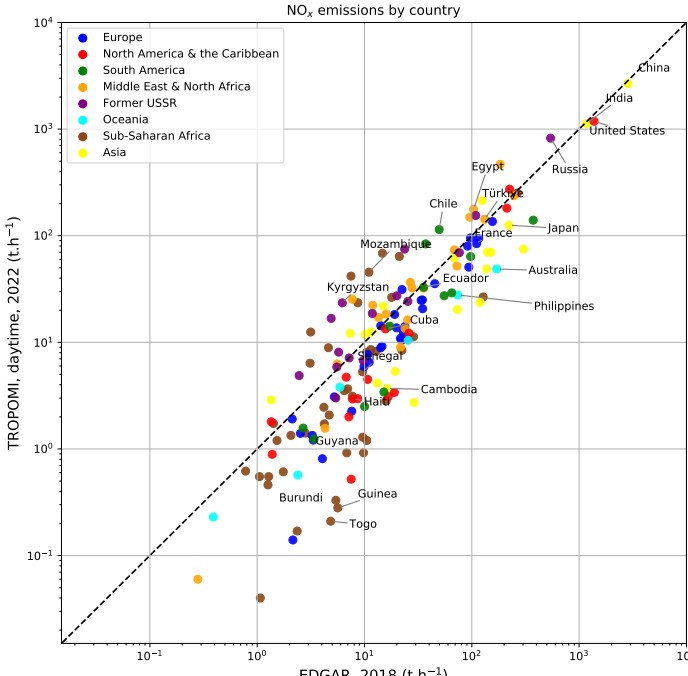

Figure 8: Comparison between TROPOMI-inferred daytime NO$_x$ emissions for 2022 (expressed as NO$_2$) and mean emissions from EDGARv6.1 in 2018 for all countries, classified by macro-regions.

The other countries for which the difference between our TROPOMI-inferred emissions and EDGAR estimates is significant are low-income countries. For such countries, it is possible that most sources are too small to be detected

with our method, resulting in an under-estimation of emissions. Countries for which TROPOMI-inferred estimates are lower than EDGAR estimates by more than an order of magnitude are Guinea-Bissau, Equatorial Guinea, Togo, Guinea, Gabon, Montenegro, El Salvador, Liberia, Ivory Coast and Myanmar. Of all these countries, only Montenegro and Gabon are not considered low-income countries. Conversely, although no country has TROPOMI-inferred estimates higher than EDGAR estimates by more than an order of magnitude, notable biases exist. In this respect, largest differences are found in central Asia (ratios of 3.8 for Kyrgyzstan, 3.2 for Uzbekistan, 3.4 for Tajikistan), in central and southern Africa (ratios of 5.6 for Zambia, 4.7 for Democratic Republic of Congo, 4.0 for Eswatini, 4.1 for Mozambique, 3.0 for Angola), and Yemen (ratio of 3.2). For these countries, it is also possible that the corresponding EDGAR estimates are imprecise, due to the incomplete or outdated nature of the reported sources in these countries. The presence of many sub-Saharan African countries with extreme differences between TROPOMI-inferred and EDGAR estimates explains why the macro-region has the highest mean absolute error despite having the lowest total relative bias.

At the global scale, our TROPOMI-inferred daytime emissions for all considered countries (i.e., excluding emissions which take place at sea and smaller countries) reach a total value of 11,168 t.h$^{-1}$. This value is consistent with that of EDGAR at 12,254 t.h$^{-1}$, i.e around 107 Mt per year, and lower than the value of 123 Mt calculated by Stocker (2014) for global anthropogenic emissions in 2000 (which include shipping and aircraft emissions). If the lower value can be interpreted as a reduction of NO$_x$ emissions between the two dates, it is also possible that our emissions are under-estimated due to biased-low columns in the TROPOMI NO$_2$ operational product (Verhoelst et al., 2021). We detail this uncertainty in Section 4.2. We should also note that our TROPOMI-inferred emissions only represent daytime emissions taken around 13:30 LT for each pixel, which are generally lower during mid-day than other times of the day, when pollution peaks in the early morning and late afternoon are reported for traffic in most cities (Menut et al., 2012; Goldberg et al., 2019). For the power sector, emissions at 13:30 are generally similar to the daily mean for power plants used for electricity baseload, but for power plants whose purpose is to meet peak demand, the mid-day emissions can largely differ from the daily mean. For other sectors such as cement, it is difficult to assess whether mid-day emissions are higher than the daily average, since cement production can be driven by factors that are more irregular than those driving power generation or traffic.

| Region | TROPOMI 2022 (t.h$^{-1}$) | EDGAR 2018 (t.h$^{-1}$) | Relative bias VS EDGAR (weighted average) | Mean absolute error VS EDGAR (unweighted average) |
|---|---|---|---|---|
| Sub-Saharan Africa | 660 | 712 | -7.4 % | 95.0% |
| Rest of Asia | 4584 | 5482 | -16.4 % | 54.4% |
| Europe | 830 | 1092 | -24.0 % | 40.3% |
| Middle East & North Africa | 1531 | 1125 | 36.1 % | 52.1% |
| North America & the Caribbean | 1690 | 1944 | -13.1 % | 48.8% |
| Oceania | 92 | 282 | -67.5 % | 62.9% |
| South America | 514 | 762 | -32.5 % | 61.8% |
| Former USSR | 1268 | 856 | 48.2 % | 77.4% |
| **Total** | **11168** | **12254** | **-8.8 %** | **64.7%** |

Table 3: Comparison between TROPOMI-inferred daytime NO$_x$ emissions for 2022 (expressed as NO$_2$) and mean emissions from EDGARv6.1 in 2018 for macro-regions. For each macro-region, the relative bias between total TROPOMI-inferred emissions and total EDGAR emissions is calculated. The mean absolute bias for all countries of these macro-regions is also calculated.

A source of underestimation can also come from the threshold used to filter out emissions. Here, the limit used of 0.2 Pmolecules.cm$^{-2}$.h$^{-1}$ makes it possible to eliminate residual emissions that are difficult to attribute to a source. This filtering also eliminates pixels with negative emissions that are physically impossible. Nevertheless, as negative emissions may represent NO$_x$ incorrectly distributed spatially in the transport term due to errors in the wind field, calculating the sum of emissions without the use of thresholds may be important for identifying countries and regions where the flux-divergence method is limited. In this case, total emissions reach 14,835 t.h$^{-1}$, which corresponds to an increase of 32.8% compared to the total with the application of the threshold. This estimate is therefore higher than the total EDGAR budget. The differences between the two estimates vary greatly by macro-region: it rises to 149.6% for sub-Saharan Africa, 126.7% for Oceania and 95.5% for South America. The increase is moderate in the Middle East & North Africa region and the North America & the Caribbean, (41.7% and 33.3% respectively). The difference between the two estimates is the lowest in former USSR countries, Europe and the rest of Asia (increases of 16.4%, 14.1% and 12.1% respectively). The trends observed previously regarding the reasons for the discrepancy between the TROPOMI-inferred estimates and EDGAR remain unchanged.

## 3.4 Temporal distribution and averaging size

The results presented so far concerned daytime emissions averaged on the entire year 2022. They therefore show a certain potential for mapping the sources of pollution, quantifying the corresponding emissions and characterising their type (by size and country or region). Several studies have shown the possibility to characterise a weekly cycle of $NO_x$ emissions (Stavrakou et al., 2020; Rey-Pommier et al., 2022). The use of geostationary satellites, such as the Geostationnary Environment Monitoring Spectrometer (GEMS) in East Asia (Kim et al., 2020), the Tropopheric Emissions Monitoring of Pollution (TEMPO) in North America (Zoogman et al., 2017) and Sentinel-4 (planned in June 2025) in Europe (Gulde et al., 2017), could also be used to characterise the daily cycle of emissions, leading to a significant improvement of forecasting capabilities. In our case, TROPOMI can only monitor pollution on a daily basis provided that retrievals are of high quality, and the analyses presented so far could theoretically be carried out at this temporal resolution. In the Supplementary Materials, we monitor the daily emissions of the Zaporizhia thermal power plant in Ukraine, whose activity was altered following the ongoing conflict in the country that started in February 2022. However, this type of monitoring remains rare and is more indicative of order-of-magnitude variations rather than precise emission estimates. In practice, the high sensitivity of the method to wind direction and the low signal-to-noise ratio around sources at high latitudes leads to daily emission maps that are very noisy in most cases, making it difficult to precisely monitor activity at this temporal resolution. In general, averaging is therefore required to reduce noise effects and limit the uncertainties associated to emission estimates. Here, we try to evaluate what level of averaging is necessary to limit noise effects and allow a monitoring of emissions. To this end, we consider the average daily emissions obtained for 2022 (i.e. over a maximum of 52 weeks) to be the most accurate estimate of daytime emissions. We compare this maximum averaging value with averages based on a smaller number of estimates. We compare the emissions of various emitters, calculated with an averaging period of 12, 24, 36 and 48 weeks. Figure 9 shows the results for eight urban areas, but with different latitudes, populations, levels of development and energy mixes: Ankara (Turkey), Cape Town (South Africa), Madrid (Spain), Portland (Oregon, United States), Chaguanas (Trinidad and Tobago), Saint Petersburg (Russia), Manila (Philippines) and Muscat (Oman). Portland and Manila are urban areas classified as a point sources. Figure 10 shows the results for six industrial facilities, which are all point sources located in Egypt, Australia, Mexico, Chile, India and Germany. The sources were chosen for their relative isolation from other emitters. Calculated emissions correspond to the sum of pixels around the source with densities greater than 2 Pmolecules.cm$^{-2}$.h$^{-1}$. There are two pitfalls to be avoided in this comparison:

- The first pitfall would be not to account for the seasonal cycle of emissions, which is very pronounced in some cases, and to compare chronological averages. For example, comparing the first 12 weeks of the year with the first 24 weeks of the same year would not make sense in terms of the difference with emissions averaged over the whole year, because in the first case, emissions would essentially be calculated in boreal winter, whereas in the second case, emissions would be included during spring and summer. To avoid this seasonal bias, emissions averaged over 12 weeks correspond to an average over the first week of each of the 12 months of 2022, emissions averaged over 24 weeks correspond to the first two weeks of these same 12 months, and so on.

- The second pitfall would be not to account for the weekly cycle of emissions. $NO_x$ emissions are generally lower at weekends due to a reduction in human activity in most areas (i.e. on Saturday and Sunday, or Friday and Saturday in most Arabian and North-African countries). It is therefore necessary to ensure that the proportion of weekend days and weekdays in each of the averages calculated remains the same, hence the interest in averaging by weeks (these proportions are therefore 2/7 and 5/7 respectively). We also carry out a fifth set of averaging over 24 days, i.e. 2 days per month. Since the seasonal effect (first pitfall) is generally stronger than the weekly bias (second pitfall), we therefore choose to retain the principle of selecting the same number of days in each month, even if it means making comparisons between averages where the weekend and weekday rates differ from 2/7 and 5/7. This last averaging set will be indicated as "irregular".

In the case of urban areas, the different averages uniformly distributed over time show a similarity in the emissions calculated over the time horizons for Ankara, Muscat, Cape Town, and Madrid. For these cities, the low cloud cover allows a high density of observations and optimal averaging. The 84-day averaging, and to some extent the 24-day irregular averaging, seems sufficient for monitoring emissions. This is not the case for the other urban areas studied, for which the observation density is lower, such as Manila, Saint Petersburg, and Chaguanas. For these cities, a monitoring performed with an averaging below 168 days (or even 252 days in the case of Manila) is therefore limited by noise effects. The limit-case is Portland, which has the larger difference between 84- and 336-days averagings. This is due to a limited number of observations over a small urban area which are not compensated by high-emissions like other point sources shown on Figure 10. For those point sources, similar emissions are observed after an 168-day averaging. In some cases, a 24-day averaging is also sufficient, while in others it is not. The representativeness of

emissions on such a low level of averaging should be considered with caution, as emissions from industrial plants are always more irregular than those from cities, with the exception of power stations used for baseload electricity generation. The averages over 84 days presented here represent emissions that include several days of activity and several moments of inactivity.

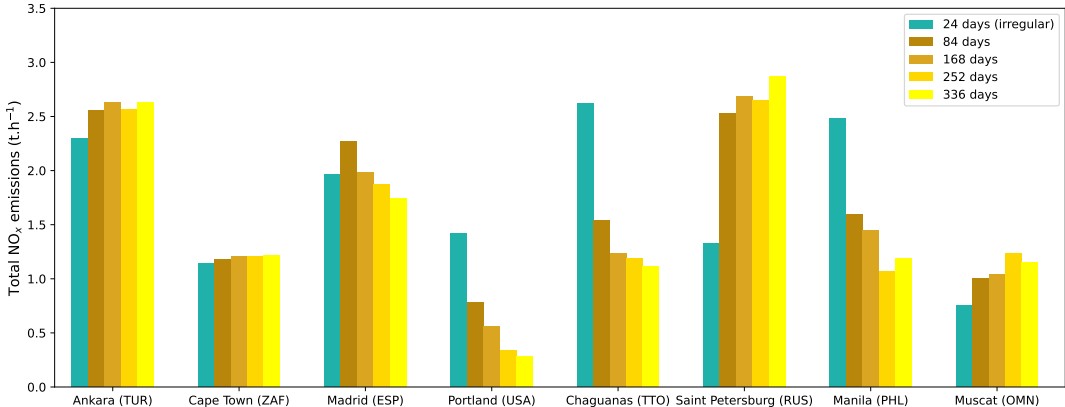

Figure 9: NO$_x$ emissions for 8 different urban areas (diffuse sources), averaged over a period of 24, 84, 168, 252 and 336 days, evenly distributed throughout the year. The proportion of weekend days and weekdays is identical in all the averaging sets except the first one of 24 days. Masses are expressed as NO$_2$.

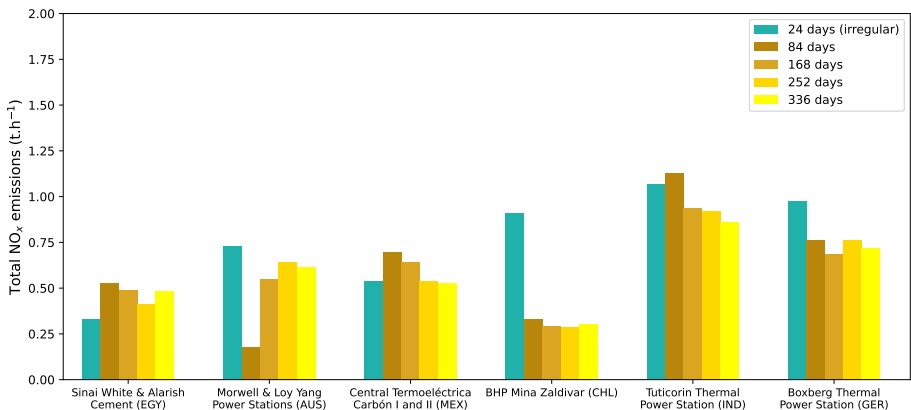

Figure 10: NO$_x$ emissions for 8 different industrial facilities (diffuse sources), averaged over a period of 24, 84, 168, 252 and 336 days, evenly distributed throughout the year. The proportion of weekend days and weekdays is identical in all the averaging sets except for the first set of 24 days. Masses are expressed as NO$_2$.

Overall, this analysis seems to indicate that tracking emissions from point sources or diffuse sources using the flux-divergence method requires an averaging effort to limit the noise obtained in the daily emissions. This averaging effort, which is made more difficult for smaller sources, increases with the density of observations, is of about a month in countries with frequent high-quality observations, but of about a year quarter in regions with low observation densities, such as tropical regions and high-latitude regions.

# 4 Uncertainties and assessment of results

## 4.1 Model uncertainties

Our top-down emissions are calculated here using a flux-divergence model, based on a simplified calculation of a transport term, a sink term and a conversion factor from NO$_2$ to NO$_x$. This simplicity reduces the computation time to calculate emissions and the dependence on external datasets, at the cost of increased model uncertainties. Here, although a "topography-wind" term has been introduced in this article to refine the transport term, the sink term remains simple and only represents the reaction between NO$_2$ and OH. While this reaction is the first contributor of

NO$_x$ loss, other sinks may be significant. For instance, organic peroxy radicals can oxidise NO$_x$ to form peroxy nitrates, making the corresponding sink important in the presence of VOCs (Stavrakou et al., 2013), especially in biomass fires. In different conditions, the formation of peroxyacetyl nitrate from NO$_2$ (Moxim et al., 1996), can also contribute to a significant share of the NO$_x$ loss. The vertical averaging of is also made simple here, and alhough the sink term varies little with the thickness of the layer within which the temperature and OH concentration are calculated (Rey-Pommier et al., 2022), this assumes the OH field is correctly represented. This assumption may be wrong if large NO$_x$ emitters are not taken into account, as this would distort the corresponding NO$_2$ field and the subsequent OH field. A possible improvement to our dataset could be to compare the columns calculated from the TROPOMI observations with the NO$_2$ column represented by CAMS and correct outliers detected from this comparison. Another refined version could infer directly the effective NO$_x$ lifetime from the NO$_2$ observations themselves, as suggested by Laughner and Cohen (2019).

Another model uncertainty comes from the calculation of the conversion of NO$_2$ production to total NO$_x$. The majority of NO$_x$ is emitted in the form of NO, which is not observed from space. A common assumption is that NO is rapidly transformed into NO$_2$ through its reaction with ozone, reaching a stationary state within a few minutes. Numerous studies (Beirle et al., 2019; de Foy and Schauer, 2022) assumed a photostationary state in typical urban conditions and used a ratio of 1.32 based on Seinfeld and Pandis (2006). Here, the values of this ratio calculated from CAMS data did not differ much from this value. However, the photostationary state is a hypothesis which is potentially not verified on the scale of a NO$_x$ source like a power plant stack. Li et al. (2023b) calculated values of this conversion ratio correlated with the combustion temperature and energy efficiency for sources in China that are highly intensive in energy such as power plants, and found a median value of 3.3. Biases in the calculation of the NO$_x$:NO$_2$ ratio can also arise in highly polluted environments, in which the Leighton relationship used to calculate this ratio is no longer valid. In particular, OH can also react with VOCs and form oxygenated VOCs. Further studies estimating this ratio at various spatial and temporal scales would thus provide a better implementation of our model.

## 4.2 Data uncertainties

The NO$_2$ column densities are the main input quantity in our estimation of NO$_x$ emissions, making the its calculation within the TROPOMI product the first element to examine when considering the data uncertainties in our estimates. Columns are calculated from measurements of solar backscattered radiation and comparison with a specific UV-Visible band using the Differential Optical Absorption Spectroscopy method, before being assimilated to derive a tropospheric vertical component. The corresponding uncertainty under polluted conditions is dominated by the sensitivity of satellite observations to air masses near the ground, and is expressed through the calculation of the tropospheric air-mass factor (AMF). To assess the significance of such effects, vertical profiles within the TROPOMI product can be replaced by any other profile information, resulting in a new retrieved tropospheric NO$_2$ column. Douros et al. (2023) replaced the *a priori* TROPOMI OFFL NO$_2$ profile by high-resolution air quality forecasts for Europe. As compared to the standard TROPOMI NO$_2$ data, this new product was found to be biased-low by 5% to 12% for most European cities. The air mass factor (AMF) itself can be replaced: for instance, Lama et al. (2022) re-calculated the AMF by replacing the tropospheric AMF of the original TROPOMI OFFL product by an AMF taken from WRF-Chem simulations. Similarly, Beirle et al. (2023) re-calculated the AMF above different emitters from the corresponding averaging kernel based on a peak profile at plume height to better reflect the distribution of NO$_2$ close to ground, which resulted in an AMF correction of about 1.61. Here, we did not perform any of such corrections, and we consider a relative uncertainty for the column of 30% (Boersma et al., 2004) for pixels corresponding to non-urbanized areas. For pixel corresponding to cities, S-5P validation activities which indicate that TROPOMI tropospheric NO$_2$ columns are systematically biased low by higher rates (Verhoelst et al., 2021), and a higher relative uncertainty of 50% is used. Such biases seem to run counter to our comparison with the catalog by Beirle et al. (2023), for which this change in sensitivity was performed but leading to emissions generally lower than ours. A more detailed analysis of the concerned emitters seems necessary to better understand the parameters that have the largest impact on the vertical sensitivity of TROPOMI retrievals and our inversion model.

Other data uncertainties can arise from other parameters that play a crucial role in the estimation of advection and chemistry effects. An accurate representation of the wind is critical to estimate the transport term correctly. For a given plume, the poor representation of wind speed leads to an under-or over-estimation of transport, but the correct orientation of positive and negative values around the source remains. However, an incorrect representation of the wind direction, such as a non-alignment with the main direction of the plume, fails to represent a correct orientation of positive and negative values. The estimation of the transport term significantly thus relies heavily on the representation of the wind angle. Higher errors are therefore expected to be high in regions having winds that vary rapidly in time, or regions with complex horizontal wind variations, such as mountainous regions. In particular,

situations where sub-grid scale-phenomena occur, not accounted for in ERA5 wind fields, might display even higher errors in the estimation of transported $NO_x$. For instance, Tehran, Iran, has an extremely complex topography, and in the calculated emissions, the transport term is particularly high compared with the sink term, with high and unrealistic negative values on large scales around the Tochal mountain immediately to the north of the city. Other megacities such as Seoul, South Korea, Jeddah, Saudi Arabia, Chittagong, Bangladesh, also exhibit unrealistically high values for the transport term. Besides, these values are not compensated by the topography-wind term, for which an inverse scale height of $X_e = 0.3$ km$^{-1}$ is used based on Sun (2022). For this term to be sufficient to compensate for the negative values observed, a higher inverse scale height would be required. Such observation is consistent with Beirle et al. (2023), who used empirical values of $X_e$ up to 2 km$^{-1}$ and reduced the amplitude of the negative patterns observed for Los Angeles, United States, Tehran, Iran, or Seoul, South Korea. Underestimations of the topography-wind term may also result from the use of a relatively coarse, postprocessed version of the topographic field, which smooths out finer-scale elevation gradients that would be better captured in the original higher-resolution GMTED data. Finally, errors in the estimation of emissions can also come from a wrong estimation of the air composition when calculating the sink term. The $NO_2$ lifetime relies heavily on the representation of the OH concentration field, which varies with $NO_x$ itself through a non-linear mechanism. An incorrect representation of the sink term can occur at the scale of a plume by not capturing this relationship due to an incorrect knowledge of emitters on the ground. This can also be due to the 0.4°×0.4° resolution of CAMS that do not always capture the $NO_2$ gradients adequately in plumes near a known emitter (Valin et al., 2011; Li et al., 2023a). For the OH concentration, a relative uncertainty of 30% has been used (Huijnen et al., 2019), representing the largest component of absolute uncertainty apart from the vertical columns. Large errors in the annual cycle of OH, and therefore in the sink term, can thus be expected. As a consequence, a wrong estimation of wind angle and OH concentration can lead to unrealistically high emissions, or even negative emissions.

# 5    Conclusion

In this study, we present a global quantification of $NO_x$ emissions by performing a mass-balance inversion based on the flux-divergence method, based. This approach offers a rapid alternative to traditional 3D inversion methods using Chemical Transport Models. The foundation of this method lies in the observation of tropospheric vertical column densities of $NO_2$ provided by TROPOMI. Our methodology incorporates several components in the calculation of emissions: a transport term driven by horizontal wind, a sink term largely driven by OH concentrations, and a topography-wind correction term. The emissions calculated represent mean daytime fluxes for the year 2022, allowing us to map emissions on a global scale. The results highlight that the primary sources of $NO_x$ emissions are industrialized and developing countries. Our emission estimates are consistent with global estimates, as well as the EDGARv6.1 inventory, though notable discrepancies are observed at the national level, particularly in former USSR countries and sub-Saharan Africa. Besides, we performed a pinpointing of emitters by distinguishing between diffuse sources, typically large metropolitan areas with extensive spatial distribution, and point sources, generally isolated industrial facilities with emissions that often exhibit a Gaussian spread. 436 diffuse sources and 323 point sources are identified. Significant uncertainties remain, especially in regions where OH is not the only source of $NO_x$ removal, regions where wind representation is inaccurate, and regions where TROPOMI data exhibit substantial biases. Nonetheless, this work demonstrates the feasibility of annual $NO_x$ emission monitoring with reduced latency and fewer mis-allocation issues compared to traditional inventories. Our approach enables the monitoring of emissions at the monthly scale in regions with high observation densities, that usually correspond to dry, mid-latitude countries. Conversely, the effect of numerical noise, combined with low-observation densities, restricts such monitoring to a higher averaging period of up to months, generally in tropical and high-latitude regions. Efforts should be made to further develop this method to provide a near-real time monitoring tool a higher temporal resolution for these regions. The results of this study were obtained from the calculation of daily $NO_x$ emissions in 2022 and their annual average.

# 6    Data availability

The monthly $NO_x$ emission maps can be accessed at `https://doi.org/10.5281/zenodo.13758447` (Rey-Pommier et al., 2025). Data is made available as emission grid maps as `.nc` files with emissions expressed in petamolecules per square centimetre per hour (Pmolecules.cm$^{-2}$.h$^{-1}$). Conversion factors to mass terms (expressed as $NO_2$, NO or N) are included. The lists of diffuse and point sources are also provided.

**Author contributions.** AR analysed the data, prepared the main software code and wrote the paper. AH improved some aspects of the code and prepared the code for Gaussian fitting. FC, PC, TC, JK and JS contributed to the improvement of the method and the interpretation of the results. All the authors read and agreed on the published version of the paper.

**Financial support.** This study has been funded by the European Union's Horizon 2020 research and innovation programme under grant agreement no. 856612 (EMME-CARE) and partially under grant agreement no. 958927 (Prototype System for a Copernicus CO2 Service (CoCO2)).

**Competing interests.** The authors declare no competing interests.

**Ethical Approval.** Not applicable.

**Consent to Participate.** Not applicable.

**Consent to Publish.** Not applicable.

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
