# Peer review of "Global gridded NOx emissions using TROPOMI observations"

_Earth System Science Data, 2024_

## Author Comment (AC1)

**Replies to comments provided by Referee #1**

Ref.: essd-2024-410
Title: **Global gridded NOx emissions using TROPOMI observations**
Author(s): **Anthony Rey-Pommier et al.**
**Earth System Science Data**
Type: Research article

We would like to thank the reviewers for their careful reading, that led to interesting comments. Reviews have been addressed in the revised manuscript and commented in this document. For a better readability, reviewer comments are highlighted in grey in this document, while answers are highlighted in light yellow.

This is an excellent study estimating NOx emissions in 2022 from TROPOMI data. I appreciate all the assumptions involved and acknowledge that there are several additional sensitivity studies that could be done but also understand most of them are beyond the scope of this manuscript. With that said, I have listed a several minor suggestions that could improve the paper and better clarify some of the unstated nuances of the work.

Major comments:

It's unclear exactly how OH is being incorporated to estimate the NO2 lifetime. Are you using surface OH concentration at the closest CAMS grid point? Or a model weighted vertical average based on the NO2 distribution? Or something more technical? If it's the former, I recommend authors perhaps looking into an improved way of inferring the OH concentration and NO2 lifetime... See Figure 1 of Laughner and Cohen https://www.science.org/doi/10.1126/science.aax6832. I would plot NO2 lifetime (calculated from CAMS OH) as a function of CAMS NO2 column data. I am assuming there will be some type of non-linear relationship that can be used to infer the NO2 lifetime when TROPOMI NO2 column data differs substantially from the CAMS column NO2 data. Ideally you'd bin by TROPOMI HCHO which I realize is beyond the scope, but maybe calculating the NO2 lifetime vs. NO2 column relationship by Koppen climate zones could be a quick work-around (which would approximately account for areas with less/more biogenic VOC emissions). This is a long way of saying that if CAMS NO2 has a large mismatch with the TROPOMI NO2 data, your assumed OH may be way off, and there could be an easy way to approximately account for these mismatches.

→ We use here averaged CAMS OH between 950 and 1000 hPa. A study done (Rey-Pommier et al., 2022) with averaging other levels showed that the impact of the thicknesses within which parameters are averaged on total emissions was low. We acknowledge that the method we use here remains basic, and bears errors if CAMS misses or mis-estimates $NO_x$ sources (this would also be seen in a mismatch between CAMS $NO_2$ and TROPOMI $NO_2$ as this comment points out). The method suggested by this comment could be an interesting improvement for a future version of the data product, and will be suggested in Section 4.1 "Uncertainties and assessment of results – Model uncertainties" in the revised version of the manuscript.

There is not enough discussion on why biomass burning emissions are not properly captured. It may be worth framing this paper as quantifying fossil-fuel related NOx emissions and purposely

screen out areas of known biomass burning NOx emissions, which appear to be particularly uncertain for a variety of reasons (as the authors correctly note).

→ We prefer to keep the title of the title as it is, because there are processes that produce $NO_x$ without involving fossil fuels (e.g. $NO_x$ is emitted in steel recycling using electricity). We also keep it because we might actually capture some biomass burning emissions – but not as properly as fossil emissions, due to factors that are discussed more in details in the revised version of the manuscript. We will however mention this uncertain nature of biomass burning emissions directly in the abstract to avoid any misguidance. We will also cite the following studies on the under-estimation of fire emissions from space: Ramo et al., 2021 (DOI: `10.1073/pnas.2011160118`); Khairoun et al., 2024 (DOI: `10.1016/j.scitotenv.2024.170599`).

In the EDGAR intercomparison, I think small mean bias shown in the "Total" value of Table 3 (i.e..., good agreement) is the product of two offsetting biases: The TROPOMI NO2 operational retrieval is biased low by ~30-50% in polluted areas/cities (Line 468), and NOx emissions are 40% larger at 13:30 local time than the 24-hour average. Therefore I don't dispute your claims in Section 3.3, but I do think that if the TROPOMI retrieval had no bias, then you would be doing an unfair comparison. More clarification should be added. I have added more references and description below.

→ We already acknowledged at the end of Section 3.3 that our comparison between estimates 13:30 LT and daily averages has limitations. We thank the reviewer for the given reference that helped us to detail this point in the revised version of the manuscript. However, a point of clarification is necessary here: in the reference provided in this commentary, mention is made of cities for which TROPOMI is biased low while emissions at 13:30 LT are higher than the daily average. In these cities, emissions are mainly transport emissions. However, in our study, we also estimate emissions from industrial facilities (power plants, cement kilns, etc). Such facilities account for a large part of the global $NO_x$ budget. In addition, they are generally located outside cities, where the TROPOMI bias is lower. Finally, their emissions at 13:30 LT are not necessarily higher than average emissions (this depends very much on the use of the industry in question: some power plants are used for baseload, while others are used primarily to meet peak demand). In conclusion, the effect of the two offsetting biases mentioned by the reviewer is indeed present for cities, but is probably less significant outside cities, in a way that depends very much on the location under consideration. More studies are needed to quantify this effect. In any case, this discussion is detailed in the revised version of the manuscript.

In Section 3.3, it would be interesting to dive a bit deeper into where there is poor agreement between EDGAR and TROPOMI. This would really demonstrate the value of TROPOMI and your method.

→ In the revised version of the manuscript, we develop where EDGAR and TROPOMI estimates disagree the most, and detail the issue of low-income countries that have small diffuse sources or low observation densities.

Detailed comments:

Line 28. A bit more nuance could be useful. You should add something along the lines of "in conjunction with sector- and country- specific NOx/CO2 ratios". There are many examples of NOx

emissions dropping rapidly but CO2 not dropping or dropping modestly. I am sure you (the authors) know this but a future reader may not.

→ The text has been changed in the revised version of the article manuscript to account for this comment.

Line 37. The authors are being generous here :-), most bottom-up datasets take 3 years to generate. Unless you know of a emission dataset developed within 1 year, I would default to saying 3 years. This would further demonstrate the utility of your method even if it take several months to process the data.

→ The text has been changed in the revised version of the article manuscript to account for this comment.

Line 82. Which levels of the wind data are used? This is important for study replication.

→ This was not precised in the first version of the manuscript – The first two pressure levels (975 hPa and 1000 hPa) are used for the wind field $\mathbf{w}$, hence the calculation of a mean horizontal wind within a layer of about 350 m above the ground. For ground wind $\mathbf{w}_g$ in the topography-correcting term, only the first pressure level (1000 hPa) is used. The text has been changed in the revised version of the article manuscript to precise this.

Line 151. Modify "minor" to "less". I also think you are misrepresenting the Beirle et al. 2019 and de Foy and Schauer 2022 studies a bit as these studies are investigating a relatively small domain over a single season or climatological pattern. A constant NO2 lifetime is not ideal, but a better assumption than if they were global studies. Please correct me if I'm wrong but I don't know of any global study assuming a constant NO2 lifetime. Beirle et al., 2023 uses a latitudinally dependent NO2 lifetime, and I agree your method of using CAMS data is much better. In short, I agree with all your sentiments here, but be careful with some of the nuance.

→ We acknowledge the two studies that are mentioned focus on a smaller domain that justifies the different computation of the lifetime. We therefore changed this text in the revised version of the article manuscript to account for this comment.

Line 202. It'd be best to move discussion in Lines 275 - 278 about wildfires to here. The missing emissions in the Amazonia suggest your method is best for estimating fossil-fuel related sources. Even though Amazonia wildfires take place for only a few months, they should probably show up more distinctly in the annual average than they currently are. Perhaps the days with the largest smoke and NOx emissions are being filtered out as clouds. Another 2-4 sentences are probably needed to discuss these nuances.

→ The general issue of wildfire emissions is now briefly investigated in the revised version of the manuscript, and the discussion has been moved where indicated by this comment.

Line 203. The sentence "Figure 3…"  should be the first sentence of the next paragraph.

→ We prefer to move this sentence in the revised version of the manuscript, but at the beginning of the paragraph where Figure 2 is introduced, as Figure 3 just consists of different zooms on Figure 2.

Line 267. I am confused by how you are counting the number of pixels in a metropolitan area. Using Baghdad as an example, I am counting maybe 30 pixels within the dotted outline in Figure 6, where does the 198 pixels value come from? And can you highlight that 198-pixel "zone" in Figure 6?

→ In the example of Baghdad, the number 198 corresponds to the number of pixels above the threshold of 2 Pmolecules.cm$^{-2}$.h$^{-1}$. It does not correspond to the number of pixels in the city core, which is only given as an indication of where emissions are the highest. For such large cities and with this threshold, the cluster generally includes the city core, the corresponding functional urban area, and highways between the city and the main industrial centres nearby. Increasing the threshold to 3 Pmolecules.cm$^{-2}$.h$^{-1}$ generally makes the distinction between the city core and the rest, as shown in Figure S4 (Supplementary Materials). The Figure below shows the 198 pixels higher than the threshold for Baghdad – The domain is slightly larger than the one in Figure 6.

[Figure]

In the revised version of the manuscript, we emphasize more on what the dotted line stands for, to avoid any confusion. However, we chose not to show explicitly the cluster zone by changing Figure 6 (like above), because we prefer to show the details of emissions on a smaller zone.

Table 1. Typo of Shanghai

→ The typo has been corrected in the revised version of the manuscript. There was also a typo for Shenz[h]en.

Lines 293 - 325. Thanks for this discussion. There is one policy-relevant question that is still unanswered in this section. From an emissions standpoint, what is the threshold point source emissions rate given a 2 Pmolec-cm-2h-1 threshold? 0.5 tons per hour? Less?

→ Of course the conversion from Pmolecules.cm$^{-2}$.h$^{-1}$ to ton.h$^{-1}$ depends on the size of the corresponding pixel (~37 kg.h$^{-1}$ at 60°N or 60°S to ~74 kg.h$^{-1}$ at the equator), which is why we prefered working with this unit (Pmolecules.cm$^{-2}$.h$^{-1}$). This comment is however relevant, and we give a range of the corresponding threshold in the revised version of the manuscript. It will be added as a comment in the metadata for the user.

Line 355. I wouldn't discount there being a real difference in Russia. How do individual cities in Russia (Moscow, St. Petersburg, etc.) compare against EDGAR?

→ We changed the word discrepancy in the revised version of the manuscript. We also compare below TROPOMI-based emissions to EDGAR for Moscow and Saint Petersburg (domain of ~1.7°×1.7° around the two cities), by summing all pixels with values above 0.2 Pmolecules.cm$^{-2}$.h$^{-1}$ in the case of TROPOMI-based estimates, as done in Section 3.3:

[Figure]

The horizontal lines represent the annual averages calculated with all daily emissions and excluding NaNs. Note that in January, February and December, no observation was taken above the domain, hence the absence of monthly estimates (for Saint Petersburg in February, a few pixels are observed but they have values below the threshold indicated above). It is also the case for some pixels in the domain in March and November. The order of magnitude is the same for both estimates, and lower emissions in summer (probably due to the lower heating demand) seem replicated. For these cities, the annual emissions therefore do not take into account the winter months, when emissions are particularly high according to the annual profile in EDGAR. The total budget for these emitters might therefore be underestimated. This situation is typical of countries where high emissions occur while the observation density is the lowest. This example will be used in the Supplementary Materials to illustrate biases for large countries with few observations during a part of the year despite having correct agreements for key emitters during the rest of the year.

Line 358. This is consistent with Ahn et al., 2023 (https://iopscience.iop.org/article/10.1088/1748-9326/acbb91) which shows something similar for CO2. I think more detail on this would be interesting and helpful. Which countries in particular show worse agreement? Are they all low-income countries and/or countries with a lot clouds? Maybe a few more of the outlier points can be labeled on Figure 8? I understand why a log-scale is used, but it is a bit deceptive as the Russia bias is probably the largest of all countries. Therefore more discussion in the text is needed.

→ The countries for which TROPOMI estimates are significantly higher than EDGAR are Kyrgyzstan, Uzbekistan, Tajikistan, Zambia, Democratic Republic of Congo, Eswatini, Mozambique, Angola, and Yemen. However, the worst agreements are found for countries where TROPOMI estimates are significantly lower than EDGAR. In Guinea-Bissau, Equatorial Guinea, Togo, Guinea, Gabon, Montenegro, El Salvador, Liberia, Ivory Coast and Myanmar, such differences are higher than an order of magnitude. These countries are countries cumulating low incomes, frequent clouds, and small size. We discuss the potential reasons for such differences in the revised version of the manuscript. We also discuss how results change when no threshold is applied when summing the emissions for countries. Finally, we added more labels in Figure 8.

Line 366-369. Urban NOx emissions at 13:30 are still ~1.4 times larger than the 24-hour average since so many nighttime hours have very low emissions: Please cite and see Figure 4a of Goldberg et al., 2019 which shows an example for New York City, United States: https://acp.copernicus.org/articles/19/1801/2019/acp-19-1801-2019.html I have seen other unpublished studies showing the temporal hourly pattern of GEOS-CF NOx emissions in many global cities look like New York City (and not Seoul). I think you have offsetting biases that are conveniently and approximately cancelling out: The TROPOMI NO2 operational retrieval is biased low by ~30-40% in polluted areas/cities (Line 468), and NOx emissions are 40% larger at 13:30 local time than the 24-hour average. Therefore I don't dispute your claims in Section 3.3, but I do think that if the TROPOMI retrieval had no bias, then you would be doing an unfair comparison. More clarification should be added.

→ We already acknowledged at the end of Section 3.3 that our comparison between estimates 13:30 LT and daily averages has limitations. We thank the reviewer for the given reference that helped us detail this point in the revised version of the manuscript. However, a point of clarification is necessary here: in the reference provided by this commentary, mention is made of cities for which TROPOMI is biased low while emissions at 13:30 LT are higher than the daily average. In these cities, emissions are mainly transport emissions. However, in our study, we also estimate emissions from industrial facilities (power plants, cement kilns, etc). Such facilities account for a large part of the global $NO_x$ budget. In addition, they are generally located outside cities, where the TROPOMI bias is lower. Finally, their emissions at 13:30 LT are not necessarily higher than average emissions (this depends very much on the use of the industry in question: some power plants are used for baseload, while others are used primarily to meet peak demand). In conclusion, the effect of the two offsetting biases mentioned by the reviewer is indeed present for cities, but is probably less significant elsewhere, in a way that depends very much on the location under consideration. More studies are needed to quantify this effect. In any case, this discussion is detailed in the revised version of the manuscript.

Line 390. Thank you for including Portland in Figure 9. First, I am assuming it is Portland Oregon, USA as there is also a Portland, Maine, USA. It is interesting that 84 days vs. 336 days of averaging shows a factor of 2 difference, whereas other cities show less variance by percent. It may be worth commenting that Portland is a relatively small city and cloudy for much of the year, so it's probably "worse case scenario" or "limit" to the type of conditions in which your method works.

→ It is indeed Portland, Oregon. The text has been changed in the revised version of the article manuscript to account for this comment.

Line 425. See prior comment. It is also a function of the size of the city/NOx source too. Large sources may only need one month of data, but smaller sources may need a full year of data.

→ The text has been changed in the revised version of the article manuscript to add the precision indicated by this comment.

---

## Author Comment (AC2)

**Replies to comments provided by Referee #2**

Ref.: essd-2024-410
Title: **Global gridded NOx emissions using TROPOMI observations**
Author(s): **Anthony Rey-Pommier et al.**
**Earth System Science Data**
Type: Research article

We would like to thank the reviewers for their careful reading, that led to interesting comments. Reviews have been addressed in the revised manuscript and commented in this document. For a better readability, reviewer comments are highlighted in grey in this document, while answers are highlighted in light yellow.

Rey-Pommier et al. present a global dataset of monthly mean gridded NOx emissions derived from TROPOMI measurements.

The authors apply a flux-divergence method which has been presented in previous studies, but, to my knowledge, has not yet been used to compile such an emission dataset on global scale. Thus, the dataset is generally of interest and matches the scope of ESSD.

However, in the current version, I see two major shortcomings that need to be resolved before publication on ESSD:

     1. The dataset contains extreme outliers, which I can only explain with some bug(s) in the processing chain, and

     2. The discussion of uncertainties misses some of the most important factors (choice of wind altitude, background correction, air mass factors).

For further details and additional comments see below.

**Methodology**

Horizontal transport

The authors do not directly state which altitude is selected for horizontal wind fields. From the context ("assumption of a stationary state and a pollution concentration close to the ground") I conclude that surface winds have been considered, which is probably not the best choice:
  - power plants etc. emit from stacks of altitudes up to some hundred meters.

  - even emissions at surface (e.g. from car exhausts) are usually rapidly mixed within the boundary layer. Thus, wind fields for a typical altitude within the boundary layer would be more appropriate, as shown in several previous studies. In any case, the authors should

- explicitly state which altitude was chosen for wind fields,

- justify that choice and

- quantify the uncertainty associated to that choice (by comparison to alternative processing with different altitude).

     → This was not precised in the first version of the manuscript – The first two pressure levels (975 hPa and 1000 hPa) are used for the wind field **w**, hence the calculation of a mean horizontal

wind within a layer of about 350 m above the ground. For ground wind $\mathbf{w}_g$ in the topography-correcting term, only the first pressure level (1000 hPa) is used. The text has been changed in the revised version of the article manuscript to precise this.

Noise

The presented emission data is very noisy at high latitudes, in particular over ocean. This is shortly mentioned in the manuscript, and the authors explain it with the low amount of available data. However, there seem to be some issues in the processing that cause quite extreme outliers: For instance, in the January data at (2348, 582), corresponding to 143.6°W, 56.8°N (northern Pacific), emissions are 20 Pmolec/cm2/h, which is a factor 10 higher than the threshold for the "high emission densities" classification in the paper. There are many more such pixel, and also many examples for negative values of the same order of magnitude. Monthly mean maps of NO2 column densities from TROPOMI are usually quite smooth, and noise of individual pixels is about 1Pmolec/cm2. If divided by lifetime, the sink term over ocean should thus be well below 1Pmolec/cm2/h. Topography is not existing over the ocean. Thus only the derivative terms could cause these high numbers. But with low column, and thus low fluxes, how can the derivative be that large?

One even more extreme case occurs at (225, 419) with emissions of -173 Pmolec/cm2/h. This corresponds to 153.8°W, 75.9°S, and might be over Antarctica, i.e. the topographic term might have issues as well.

I suspect that the derivatives and/or gridding algorithms, in combination with gaps, causes these extreme outliers. The authors should check these examples and investigate the values in the "orbit" reference frame:

   - if the values for individual TROPOMI pixels show such high values, check from which term they come from

   - if the orbit data looks reasonable, check the gridding routine.

These extreme and unexplainable outliers devalue the whole dataset. They should be identified and fixed before publication of this product on ESSD.

   → All the analyses carried out in the article are based on pixels between latitudes 65°S and 65°N, i.e. the domain shown in Figure 2. We forgot to specify this in Section 3. This choice is motivated by the systematic low-quality flag applied by TROPOMI above a certain latitude that depends on the satellite orbit and the season, as shown on the different examples below ($q_a < 0.5$ under a given along-track line).

[Figure]

With such low number of observations, mean inferred emissions rates at these latitude are very noisy. Knowing the absence of any major emitter/city above these latitudes (the only cities with more than 100,000 inhabitants above these latitudes are Murmansk and Norilsk, Russia), we expect this restriction not to influence our results.

The outlier identified by this comment is another issue. We thank the reviewer for identifying this outlier, which was not detected, and lead to the detection of an error in our routine. This error stems from an approximation in the estimation of distances between pixels, which is no longer valid in the polar regions (the code was initially developed for estimating emissions in the Middle East). This error underestimated the distance between two pixels that is used to calculate the derivatives, artificially inflating the transport term. For moderate latitudes, the inflation is negligible. However, it grows rapidy with latitude (~6% around 50° but ~94% at 75°). This error does not question the results and the trends obtained here, due to the low underestimation between 65°S and 65°N (where we carried out the analyses), but it remains problematic. We have therefore corrected the distance routine and recalculated all the emissions. All results and product data have been adjusted accordingly. The effects of this modification are the following:

- The typical noise at high latitudes is reduced (visible on Figure 2).

- Some extreme outliers disappear. The detected outlier at (225, 419) is clearly affected by the correction: its January emissions are lowered from -172.88 to -0.02 $Pmolecules.cm^{-2}.h^{-1}$ (annual emissions: -0.07 $Pmolecules.cm^{-2}.h^{-1}$). The other outlier at (2348, 582) is also affected but to a lower extent: its January emissions are lowered from 19.86 to 15.19 $Pmolecules.cm^{-2}.h^{-1}$ (annual emissions: 1.19 $Pmolecules.cm^{-2}.h^{-1}$). This second outlier is obtained because this point is observed only one time in January (out of 431 orbits). During this day (23/01), abnormally high VCD values are observed (~4 $Pmolecules.cm^{-2}$) with fast winds, leading to the calculated value.

- Depending on the sign of the transport term, many pixels have emissions that are now above or under the thresholds of 2 $Pmolecules.cm^{-2}.h^{-1}$ used to detect point and diffuse sources. As a consequence, the list of detected sources is modified. The number of point sources considered as outliers is reduced from 61 to 48. The average absolute emission change is 4.8% for diffuse and point sources, with a median at 2.7%.

- The effect above is amplified when estimating country emissions, for which a threshold of 0.2 $Pmolecules.cm^{-2}.h^{-1}$ is used. Depending on the sign of the transport term, many pixels have emissions that are now above or under this threshold. The total output of countries with very low emissions (generally in sub-Saharan Africa) is therefore changed not by the amplitude of the correction, but by the number of pixels that fall above or below the threshold. However, for high-latitude countries, the correction allows a better estimate of emissions (for instance, Russia has now emissions closer to the EDGAR estimate).

We thank the reviewer for noticing these outliers that allowed to correct and improve our NOx estimates.

Background correction

The choice for background correction is quite extreme: a full swath width of TROPOMI can cover a wide range range of conditions and rather presents a "mesoscale" than a "local" background. I would expect that changing the settings for the background correction can have strong impact on the presented results. This effect has to be investigated and quantified.

→ We chose a full swath width to ensure the background is calculated on a high number of pixels, which would not be the case in cloudy regions. The comment made here remains valid: using such a large region covers several climates. However, as the across-track direction is closer to the east-west axis than the north-south axis, the variations are less marked there than in the along-track direction, where climatic variations can be strong. To quantify the optimal trade-off between those two effects, below is displayed the background calculated at six different pixels in January 2022 when using: 1/ the total satellite swath (~700 km × 2400 km); 2/ a third of it (~700 km × 800 km); 3/ a tenth of it (~250 km × 700 km), centered around the pixel of interest. The pixels have been chosen in order to have examples of rural/empty areas (in Iraq and Mozambique), small cities (Saint-Louis, Senegal and Joinville, Brazil) and the core of large cities (Marseille, France and Sydney, Australia) in the two hemispheres.

[Figure]

Small differences appear between the three estimates. They generally remain below ~0.4 Pmolecules.cm$^{-2}$, and the use of one particular background has therefore a negligible impact on emissions estimated for powerful emitters (first row), for which VCDs are higher than the background by an order of magnitude. The effect is also low for rural/empty areas (third row), where the three backgrounds roughly equal the value of the point of interest, leading to a zero or near-zero VCD in all three cases. However, the choice of the background has an impact for medium-size emitters (second row): for such emitters, the background calculated on the ~250 km

× 700 km zone can be similar to the VCDs around the emitter. This happens when a lot of low-quality observations are located within the background calculation domain, but not near the point of interest. In this case, some pixels with intermediate pollution levels can be below the 30th percentile or the remaining observed pixels, therefore increasing the background to levels close to the medium-size emitter. This effect is less pronounced when the total swath is used. We thus prefer to use the total satellite swath for the background calculation, despite the associated drawbacks highlighted by this comment. In the revised version of the manuscript, we briefly precise this reasoning.

What is the reason for including a far larger distance across-track than along track?

→ When selecting pixels in the along-track (roughly north-south) direction, one must consider the different origins of background $NO_x$ due to different climatic and geographical conditions. For instance, in a region of interest like Cyprus, extending too far in the north-south direction could result in mixing pixels from arid areas (e.g. Egypt) with more temperate or humid regions (e.g. Ukraine). These areas have distinct $NO_x$ backgrounds originating from different sources, such as soil emissions and human activities, which could skew the analysis if mixed. Conversely, variations in $NO_x$ levels are generally less pronounced in the across-track (roughly east-west) direction, making it more suitable to extend the pixel selection in this direction. However, selecting a large extent across the satellite track also carries the risk of including measurements taken at different local times, which could be problematic as $NO_x$ levels vary with OH which has a strong diurnal cycle. In this specific case, the risk of temporal inconsistency is minimal because the satellite swath has a relatively small longitudinal span, and the overpass occurs around midday when diurnal variations are less pronounced. Therefore, extending the selection in the across-track direction provides a better representation of the background $NO_x$ levels. In the revised version of the manuscript, we briefly precise this reasoning.

The choice of the 1st tercile for defining the background implies that 1/3 of all considered TROPOMI pixels have corrected columns < 0. This should be mentioned, and this is also one of the reasons for negative emissions.

→ There might be a misunderstanding here. When a TROPOMI pixel has a vertical column lower than the calculated background, the corresponding corrected column is reduced to zero: the field of corrected columns is never negative. To make sure this aspect is understood correctly, we detail this in the revised version of the manuscript.

Negative emissions

The dataset includes negative values for numerous pixels. This is shortly discussed in the manuscript, but this discussion should be extended. In particular, the authors should provide a recommendation to the user how to treat negative emissions in potential applications. In my point of view, it makes sense to keep the negative results in the data product, even if unphysical, as the alternative (skipping or setting them to 0) would bias high the overall mean. But this is exactly what happened in the spatial integration of country/regional emissions, which only considers pixels above a (positive) threshold. This might be one reason for the results being systematically higher than those from previous studies (even though no air mass factor correction was applied). The authors should discuss this aspect, and should provide information about how the derived

emissions depend on the selection of pixels, and how they would look like if also the negative emissions would be considered in the spatial integral.

→ This point is particularly relevant for countries with no powerful sources or with low observation densities, leading to a majority of noisy pixels. For other countries however (industrialized countries with typical or high observation densities), the impact is relatively low. We discuss this aspect in the revised version of the manuscript and provide comparisons of country emissions when the positive threshold of 0.2 Pmolecules.cm$^{-2}$.h$^{-1}$ is used (excluding noise but also high negative values), and when no threshold is applied. In the revised version of the manuscript, a complete paragraph is dedicated to this comparison after Figure 8 and Table 3.

Air mass factors

The authors just took the operational product without any correction of the air mass factor (AMF). This is problematic, as the AMF, and also its stated uncertainty, refer to the full tropospheric column, while the authors performed a background correction with the aim to only consider near-surface pollution. For this, the operational AMFs are not appropriate, and are systematically too high (and thus Omega' is biased low). This systematic effect cannot be described by a 30% uncertainty (in both directions).

→ Verhoelst et al., 2021 indicate that the effect of the AMF leads to tropospheric NO$_2$ columns that are systematically biased low by about 30%–50%, but only over cities. This percentage is lower elsewhere. We used a 30% uncertainty in our estimates of emissions for entire countries, where emissions do not take place only in cities. However, as the reviewer describes here, a higher uncertainty must be used for estimates of cities only (and, to a lesser extent, large industrial facilities that have similar albedo values). We now make this distinction in the revised version of the manuscript.

Uncertainties
The section on uncertainties needs to be extended with discussions of the aspects listed above. The presented agreement to EDGAR is quite good, but due to the issues discussed above, there might be some of the systematic effects compensate each other.

→ In the revised version of the manuscript, we discuss the possibility of offseting biases that can cancel out following a comment from *Anonymous Reviewer #1.* Goldberg et al. (2019) have shown that emissions at 13:30 LT are higher than the daily average. In these cities, emissions are mainly transport emissions. However, in our study, we also estimate emissions from industrial facilities (power plants, cement kilns, etc). Such facilities account for a large part of the global NO$_x$ budget. In addition, they are generally located outside cities, where the TROPOMI bias is weaker. Finally, their emissions at 13:30 LT are not necessarily higher than average emissions (this depends very much on the use of the industry in question: some power plants are used for baseload, while others are used primarily to meet peak demand). In conclusion, the effect of the two offsetting biases mentioned by the reviewer is indeed present for cities, but is probably less significant outside cities, in a way that depends very much on the location under consideration. This discussion is detailed in Section 3.3 of the revised version of the manuscript.

Purpose

In the abstract, the authors claim that "this dataset is designed to be updated with a low latency to help policymakers monitor emissions". I think that this is an important aspect, as monitoring of changing emissions in timeseries might even work with the remaining high uncertainties. However, this aspect is not discussed any further in the manuscript. It would help a lot to support this argument if the authors could find e.g. a power plant that has been switched off in 2022 and show the corresponding time series of monthly mean emissions.

→ This review is interesting as it suggests to use a precise example to support the scope of our article. It is highly difficult to find information about a power plant that:
- has high emissions to illustrate the method properly,
- has a high density of observations,
- is isolated from other industrial activities and cities,
- whose activity was drastically altered in 2022 (in most cases, it is only a few turbines that are turned off for maintenance), and
- whose activity was precisely reported in global databases or covered in the press.

In the revised version of the manuscript, we illustrate this with the example of the Zaporizhia thermal power plant in Ukraine (not to be confused with the neighbouring nuclear power station), whose activity was altered following the ongoing conflict in the country. This power plant ticks most of the boxes above, but the interpretation of the monthly time series remains uncertain due to low observation densities in January and a lack of knowledge of the power plant activity in the fall. For this reason, we prefer to detail this example in the Supplementary Materials and briefly mention it in the revised version of the manuscript.

**Dataset**

- Unit

I understand the origin of the unit used (Pmolecules.cm−2.h−1) as this is just the commonly used unit for NO2 column densities (Pmolecules.cm−2) divided by a lifetime. However, the presented dataset provides global emissions, which is of high interest for communities beyond those familiar with satellite NO2 products. Thus, the unit should be modified to an (SI) unit commonly used, like kg/m2/s. At least, a conversion factor needs to be provided in manuscript and data product metadata.

→ To our knowledge, there is no global norm on which $NO_x$ must be expressed. Some studies count $NO_x$ in $\mu g$, $kg$ or $t$ (tons), but as if all $NO_x$ molecules were $NO_2$, some others use NO, and some only count nitrogen atoms. The confusion of such units is the source of many mistakes in articles when this is not clearly specified. Here, we wanted to avoid confusion for future users by using the *Pmolecules* unit, and only convert to tons per hour when presenting emissions of power plants, cities and countries in Sections 3.2 to 3.4. It is also a quite convenient unit since it allows to express pixel emissions usually between 0.5 and 10 $Pmolecules.cm^{-2}.h^{-1}$, which are easier ranges to work with. However, we acknowledge that the interest of this article goes beyond the scope of Earth Observation. In the revised data product, we specify the unit and provide six conversion factors (in $kg.km^{-2}.h^{-1}$ and $\mu g.m^{-2}.s^{-1}$, counted as $NO_2$, NO and N).

- NC files

The coordinates should directly be lat and lon. There is no need for a "lat_grid" index. The unit "degrees_north" for the grid *index* makes no sense. When having lat and lon as coordinates, there is no need for additional "latitude_data". I propose to add the grid pixel area (1d, latitude dimension only) such that the user can simply convert the emission densities to total emissions for each grid pixel, which simplifies spatial integration.

→ We modified the data product accordingly, following the advice in this comment: each monthly file of the dataset has now three variables: longitude (1D, 5760 elements), latitude (1D, 2880 elements), and NOx_emissions (2D, 2880×5760 elements).

- Annual mean

As the figures in the paper often display annual means, also an annual mean data file should be provided next to the monthly means.

→ A mean data file has been added in the new version of the data product to account for this comment (see comment above).

- Negative values

The negative values should not be skipped in the dataset. But I would propose to add a disclaimer to the metadata of each file explaining the reason for the occurence of negative data.

→ Negative values are included in the dataset. A "readme" file is added to the new version of the data product, explaining briefly how the emissions are estimated, the reason for negative data, and a reference to the article for more information.

**Minor comments**

Line 64: A reference to TROPOMI (e.g. Veefkind) should be added.

→ This is done in the revised version of the manuscript.

Line 68: This high spatial resolution only holds for nadir pixels.

→ We changed "3.5×5.5 km$^2$ since 6 August 2019" to "up to 3.5×5.5 km$^2$ since 6 August 2019" in the revised version of the manuscript.

Line 92: Provide information about the spatial resolution of GMTED2010. How exactly is this data "regridded on the TROPOMI grid"? I assume that one TROPOMI pixel covers many GMTED2010 pixels, and simple interpolation of GMTED2010 data to the TROPOMI center pixel coordinates would not be appropriate. Instead, all covered GMTED2010 pixels should be averaged.

→ The resolution of GMTED2010 (0.0625°, `https://temis.nl/data/gmted2010/`) is now being indicated in the Section 2.2.2 of the revised version of the manuscript. A TROPOMI pixel generally covers only one GMTED2010 pixel. It was also corrected on Figure 1.

Line 110: "resulting in a purely horizontal calculation of emissions":  The considered transport is "purely horizontal". Emissions are calculated as sum of divergence and sink term (proportional to the column).

→ It was meant here that the model was using 2D arrays in its calculation of emissions. The words "purely horizontal" have thus been changed to "purely 2D" in the revised version of the manuscript.

Line 113: Of course the approach requires input data like wind fields which have uncertainties. But on top, the assumptions (stationary state and a pollution concentration close to the ground) are probably just wrong afar from strong local emission sources.

→ This precision has been added in the revised version of the article manuscript.

Line 130: Note that even without NOx from lightning and soil emissions, there would be a free tropospheric background from long range transport of NOy & PAN.

→ This precision has been added in the revised version of the article manuscript, with a reference (Zhai et al., 2024).

Line 139: This reads as if the enhanced NO2 over ship tracks is caused by lightning NOx, but it is primarily caused by direct ship emissions.

→ Shipping emissions enhance the $NO_2$ column but also influence lightning activity that creates $NO_x$. To avoid any conclusion, we changed the sentence highlighted by this comment.

Line 141: Sink term: k[OH] needs to be multiplied with Omega.

→ This typo has been corrected in the revised version of the manuscript.

Line 143: Please provide more specific information, e.g. global maps of NOx lifetimes from CAMS for Jan and Jul, which might be added to the Supplement.

→ The sentence relative to this comment has been made more specific. Two global maps of $NO_x$ lifetimes (DJF and JJA) have also been added in the revised version of the Supplement.

Line 150: In particular, CAMS cannot resolve the extreme conditions within power plant plumes. This should at least be mentioned.

→ This text has been added in the revised version of the article manuscript to account for this comment.

Equation 4: I think that the topographic correction must be determined from Omega, not Omega', as it describes the change of the *background* column when blown over a inhomogenous terrain. With Omega', the effect will be underestimated.

Equation 5:

- is the divergence calculated for Omega or for Omega'?

- k[OH] needs to be multiplied with Omega.

→ These typos have been corrected in the revised version of the manuscript.

Line 239: Tehran was discussed in Beirle et al., 2023 (Fig. A1 therein) as example for the benefit of the topographic correction: without correction, emissions were negative, but with topographic correction, maps were far more plausible. I suspect that the topographic correction does not work properly here, which might be due to the way of interpolation of the GMTED2010 data to TROPOMI grid and/or the choice of Omega' instead of Omega in Eq. 4. Please check.

→ The reasons why Beirle et al. (2023) are more plausible is that they studied Los Angeles, USA, Tehran, Iran, Seoul, South Korea, and the Shanxi province in China with three different empirical scale heights: H = 0.5 km, 0.666 km and 1 km. This differs from our study, which uses a scale height of H = 3.333 km ($X_e$ factor = 1/H = 0.3 $km^{-1}$). Our value of $X_e$ (or H) is based on Sun, 2022 (Supplement of `https://doi.org/10.1029/2022GL101102`), for which H = $1/X_e$ = 3 km on average over the year, based on the US mountainous cells above the US. Here, we have preferred to use this last order of magnitude as it represents the average value of a times series deduced by fits, rather than the empirical values of Beirle et al. (2023), while admitting that using smaller values of H seems to solve the problem better for the cities mentioned above. Note that the choice of $\Omega$ instead of $\Omega'$ would not solve the issue, as the difference between the two is very low in polluted areas (generally $\Omega-\Omega' \ll \Omega$). We will however mention the work of Beirle et al. (2023) for this aspect in the dedicated section in the revised version of the manuscript.

Fig. 4: The extreme outlier of -173 Pmolec/cm2/h occurs in January, while Fig. 4 refers to annual means. But even if values would be close to 0 for the other months, the annual mean at that pixel would be about -14 Pmolec/cm2/h, i.e. there should be at least one "negative" pixel with absolute value > 10. Why does this not show up in Fig. 4?

→ This is because this pixel is outside the interval [65°S, 65°N], precisely (-75.90625, -153.78125) in Antarctica. As explained above, all the analyses in this article were carried out for this domain (corresponding to the area shown in Figure 2). Note: for this pixel, the only months for which there is at least one observation are January, February, March and December, with estimated emissions of -172.88, 2.21, 0.01 and -63.89 Pmolecules.$cm^{-2}.h^{-1}$, i.e. a final average value of -58.89 Pmolecules.$cm^{-2}.h^{-1}$ for 2022.

---

## Author Response (AR2)

**Author's tracked changes for the updated version of:**

Ref.: essd-2024-410 Title: **Global gridded NOx emissions using TROPOMI observations** Author(s): **Anthony Rey-Pommier, Alexandre Héraud, Frédéric Chevallier, Philippe Ciais, Theodoros Christoudias, Jonilda Kushta and Jean Sciare Earth System Science Data** Type: Research article
* * *
**Section 2.1.3**

**Removed:**

For computing altitude gradients, we use the Global Multi-resolution Terrain Elevation Data (GMTED2010, Danielson and Gesch (2011)) with a 0.0625°×0.0625° resolution. Elevation data is re-gridded on the TROPOMI grid, before calculation of the corresponding gradient to derive a corrective "topography-wind" value that is detailed in Section 2.2.2.

**Added:**

For computing altitude gradients, we use the Global Multi-resolution Terrain Elevation Data (GMTED2010, Danielson and Gesch (2011)) in its 0.0625°×0.0625° resolution version provided by the TEMIS data portal (https://temis.nl/data/gmted2010/). This version is derived from the original higher-resolution GMTED product (available at 30, 15, and 7.5 arc-seconds) to conveniently match coarser spatial scales. Elevation data is re-gridded on the TROPOMI grid, before calculation of the corresponding gradient to derive a corrective "topography-wind" value that is detailed in Section 2.2.2.

**Figure 6**

The Figure has been updated because negative pixels were filtered out by mistake when generating the image. The new version does not fail on this point, and corresponds more to the corresponding Figure in the original version of the manuscript submitted in September 2024.

\_\_\_\_\_

\_\_\_\_\_

**Section 4.2**

Added:

Underestimations of the topography-wind term may also result from the use of a relatively coarse, postprocessed version of the topographic field, which smooths out finer-scale elevation gradients that would be better captured in the original higher-resolution GMTED data.

**References**

All references are now identified (DOI, ISBN, etc.). The same has been done in the Supplementary Materials of the article.